# Efficient Test-time Inference for Generative Planning Models with OCL Search

**Robert Gieselmann** [1]  **Mihai Samson** [1]  **Federico Pecora** [2]  **Jeremy L. Wyatt** [1]

## Abstract

Generative models have emerged as a powerful paradigm for AI planning, yet their performance remains constrained by the training data distribution. One approach is to improve generated solutions during inference by scaling test-time compute. A more efficient alternative is to optimize the inference process itself. In this paper, we show that a modified version of a classical Open-Closed List (OCL) search provides just such an efficient inference procedure. Our algorithm synergizes two learned components: a generative model that performs fast rollouts from intermediate states and a heuristic model that prioritizes among candidate reasoning paths. Key contributions include novel exploration control mechanisms and integration of learned models within the OCL framework. Across multiple combinatorial planning domains, our approach outperforms both neurosymbolic search baselines and classical solvers in computational efficiency and solution quality.

## 1. Introduction

Automated planning seeks action sequences that transform an initial state into one satisfying goal conditions. Deep generative models have emerged as a promising paradigm for plan generation, offering fast synthesis across diverse domains (Rossetti et al., 2024b). However, solution quality is bounded by the training data, and collecting large-scale optimal datasets is often infeasible; even seemingly simple domains like Blocksworld are NP-hard to solve optimally (Slaney & Thiébaux, 2001).

An alternative is to spend additional compute at inference time. Best-of-N sampling (Stiennon et al., 2020) repeatedly queries a generative model and returns the best generation,

but does not balance exploration and exploitation to find better solutions at lower cost. Search algorithms like Monte Carlo Tree Search (MCTS) have become popular for test-time compute in formal reasoning domains such as theorem proving (Chen et al., 2024; Hubert et al., 2025) or code generation (Antoniades et al., 2025). However, MCTS methods typically employ UCT-style selection (e.g., PUCT; Rosin, 2011) with exploration incentives growing with parent visitation counts. Because every traversal originates at the root, deeper nodes are visited less frequently, often yielding wide but shallow trees. This depth imbalance is problematic for long-horizon planning, where critical decisions deep in the tree remain underexplored.

Open-Closed List (OCL) algorithms (Hart et al., 1968; Valenzano & Xie, 2016) like A* select nodes globally and underpin state-of-the-art symbolic planners (Helmert, 2003). However, unlike MCTS where generative models can serve as rollout policies, standard OCL provides no mechanism for fast candidate generation nor does it compensate for inadmissible heuristics. When trained on suboptimal demonstrations, learned heuristics overestimate cost-to-go, disproportionately inflating estimates for states far from the goal and causing greedy behavior that favors near-goal nodes. Moreover, querying generative models is computationally expensive, necessitating selective invocation.

We present OCLGEN, a compute-efficient test-time search algorithm for autoregressive planning models that substantially improves plan quality. Our approach addresses these challenges through three innovations: (1) *depth-partitioned selection* maintains separate open lists per depth level, ensuring balanced exploration despite overestimating heuristics; (2) *truncated rollouts with adaptive expansion* uses the generative model for fast multi-step synthesis while branching at low-confidence decision points; and (3) *distributional heuristic estimation* ranks nodes by a lower-percentile cost-to-go estimate, targeting their best attainable outcome.

Experiments on four classical planning domains demonstrate that OCLGEN converges to shorter plans significantly faster than baselines. On problems with known optimal solutions, our method achieves 87.3% optimality across domains, compared to 49.8% for MCTS given the same compute budget. OCLGEN also provides an effective foun-

[1]Amazon [2]Work done while at Amazon. Correspondence to: Robert Gieselmann <robgie@amazon.com>.

dation for iterative self-improvement: after three refinement iterations, our method achieves 100% optimal plans on Blocksworld and 94.7% on Sokoban, compared to 13.5% and 51.3%, respectively, for the base model with Best-of-N sampling given the same runtime budget. In summary, our contributions include:

- A modernized OCL framework with autoregressive models for fast rollout generation.
- A distributional approach for deploying overestimating heuristics in search.
- Comprehensive evaluation across four domains with ablations validating each design choice.
- Demonstration of OCLGEN's effectiveness within a recursive self-improvement framework.

## 2. Related Work

**Generative Models for AI Planning**   Deep generative models have emerged as a promising approach for planning by learning to generate solutions from collections of solved instances. Recent work has explored both pre-trained large language models (LLMs) and smaller domain-specialized transformers for plan generation. While LLMs can be fine-tuned for planning tasks (Pallagani et al., 2023), concerns about inference costs and reliability have motivated training smaller transformers from scratch on planning data. These approaches typically formulate planning as autoregressive sequence generation, predicting plans token by token. For example, PlanGPT (Rossetti et al., 2024b) trains a GPT2 model on plans from a domain-independent planner by splitting individual actions into sequences of operator name and object tokens. Extensions include integrating action validators during generation (Rossetti et al., 2024a), repairing invalid plans via local search (Tummolo et al., 2024), and leveraging symmetries for improved generalization (Fritzsche et al., 2025). Our work adopts the model architecture and tokenization scheme from (Rossetti et al., 2024b) and contributes an efficient inference algorithm.

**Test-time Search**   Test-time inference in generative models is increasingly framed as a search over candidate solutions or reasoning paths. Recent frameworks include Tree of Thoughts (Yao et al., 2023), which applies breadth-first search or depth-first search to deliberative reasoning, and rStar (Guan et al., 2025), which leverages MCTS for multi-step reasoning. In formal domains, DeepSeek-Prover-V1.5 (Xin et al., 2025) integrates MCTS with proof assistant feedback, while PG-TD (Zhang et al., 2023) combines AlphaZero-style search with test case execution for verified code generation. While most of these methods target general LLM reasoning, in this work we focus on smaller, domain-specialized planning models, where the same search principles can substantially improve output quality.

**Self-Improvement via Search**   Test-time search can serve as a standalone inference procedure or underpin recursive self-improvement, where search-generated solutions provide supervision for retraining. Combining search and learning in a self-improvement loop dates back to early checkers programs (Samuel, 1959), and has since been applied to classical planning via heuristic search with learned neural heuristics (Jabbari Arfaee et al., 2011; Groshev et al., 2018). The idea gained widespread attention with AlphaGo (Silver et al., 2016) and AlphaZero (Silver et al., 2017), which coupled MCTS with deep networks trained via self-play. Similarly, the Expert Iteration (ExIt) framework (Anthony et al., 2017) formalizes this loop, where an *expert* search iteratively supervises an *apprentice* policy. Analogous frameworks have recently been applied to fine-tuning transformers for reasoning (Lehnert et al., 2024; Zhang et al., 2024; Chen et al., 2024), theorem proving (Xin et al., 2025), and multi-step reasoning (Guan et al., 2025). The generative planning framework in Gieselmann et al. (2026) combines Best-of-N sampling with graph processing to refine a transformer in a recursive loop that iterates between data curation and model finetuning. We build on this work, focusing on efficient test-time search which further accelerates self-improvement.

## 3. Preliminaries

### 3.1. Classical Planning in PDDL

We focus on classical single-agent planning in deterministic, fully observable, discrete environments. A planning problem is specified by a finite set of objects $\mathcal{O}$ and a finite set of predicates $\mathcal{P}$, where each predicate $p \in \mathcal{P}$ has an associated arity $\mathrm{arity}(p)$. Grounded atoms are formed by instantiating predicates with objects: $\mathcal{F} = \bigcup_{p \in \mathcal{P}} \left\{ p(o_1, \ldots, o_{\mathrm{arity}(p)}) \mid o_1, \ldots, o_{\mathrm{arity}(p)} \in \mathcal{O} \right\}$. A state $s \in \mathcal{S} = 2^{\mathcal{F}}$ represents the set of atoms that hold true in a given world configuration. Actions $a \in \mathcal{A}$ are grounded operators $a = \langle \mathrm{pre}(a), \mathrm{add}(a), \mathrm{del}(a) \rangle$, where preconditions $\mathrm{pre}(a) \subseteq \mathcal{F}$, add effects $\mathrm{add}(a) \subseteq \mathcal{F}$, and delete effects $\mathrm{del}(a) \subseteq \mathcal{F}$ determine applicability and outcome. An action $a$ is applicable in state $s$ when $\mathrm{pre}(a) \subseteq s$, yielding successor state $F(s, a) = (s \setminus \mathrm{del}(a)) \cup \mathrm{add}(a)$. Goals $s_g \in \mathcal{G} \subseteq 2^{\mathcal{F}}$ are partial state descriptions. A state $s$ satisfies goal $s_g$ if $s_g \subseteq s$. A plan $\tau = (a_0, \ldots, a_{T-1})$ is valid if each action $a_t$ is applicable in $s_t$, and executing the sequence from $s_0$ via $s_{t+1} = F(s_t, a_t)$ reaches a final state $s_T$ satisfying the goal. In this work, we define optimality as minimum plan length.

The Planning Domain Definition Language (PDDL) (McDermott et al., 1998; Fox & Long, 2003) serves as the standard formalism in AI planning, underpinning benchmarks such as the International Planning Competition (Vallati et al., 2015). A PDDL domain defines predicates, typed objects, and operators specified through preconditions and effects.

Individual problem instances declare concrete objects, an initial state, and goal conditions. We use PDDL as the formal language to formulate problems, states, and plans while leveraging its structure to tokenize planning problems for autoregressive sequence generation.

## 3.2. Autoregressive Planning Models

We investigate test-time inference for generative models $\pi_\theta$ that produce distributions over actions conditioned on an initial state $s_0 \in \mathcal{S}$, a goal specification $s_g \in \mathcal{G}$, and the sequence of previous actions $a_{<t} \in \mathcal{A}^t$, i.e., $\pi_\theta(a_t \mid s_0, s_g, a_{<t})$. Following Rossetti et al. (2024b), we model $\pi_\theta$ as a decoder-only transformer (Radford et al., 2019) that represents states and actions by tokenizing PDDL descriptions of predicates, operator names and objects. This representation enables the model to navigate the combinatorially large space of grounded actions using a compact, fixed-size vocabulary. Actions are split into tokens and generated autoregressively.

To illustrate, consider the Blocksworld domain — a classical planning benchmark where the objective is to rearrange a set of blocks (e.g., b1, b2, b3) into a specified goal configuration using four operators: pickup and putdown for moving blocks to and from the table, and stack and unstack for placing or removing blocks on top of one another. In this domain, the grounded action unstack b1 b2 (removing block b1 from atop block b2) is encoded as a sequence comprising the operator token unstack followed by argument tokens b1 and b2. At each generation step, the newly predicted token is concatenated to the input sequence for the subsequent forward pass. The vocabulary is derived directly from PDDL domain and problem files, comprising predicate names, object identifiers, type declarations, and special delimiter tokens. A domain-specific upper bound on the number of objects ensures bounded representations.

# 4. Efficient Test-time Search for Autoregressive Planning

Using the generative model $\pi_\theta$ defined in Section 3.2, we develop an efficient test-time search algorithm. Our objective is to minimize plan length while generalizing across a distribution of problem instances $P_{\mathcal{S} \times \mathcal{G}}$. For each planning domain, a model $\pi_\theta$ is trained via supervised learning on $\mathcal{D}_{\text{train}}$, a dataset of suboptimal plans generated by running a symbolic solver on a set of instances drawn from $P_{\mathcal{S} \times \mathcal{G}}$.

## 4.1. OCL Search with Generative Models

We introduce OCLGEN, an efficient test-time search algorithm for generative planning models built on the Open-Closed List (OCL) framework (Hart et al., 1968; Valenzano & Xie, 2016). This framework represents a general class of

graph search algorithms that includes, e.g., $A^*$. OCL search maintains an open list of frontier nodes pending exploration and a closed list that tracks already-visited states to prevent redundant expansions. At each iteration, a node $n$ is selected from the open list $\mathcal{O}$ according to a priority function $f(n) = g(n) + h(n)$, where $g(n)$ denotes the incurred path cost and $h(n)$ the heuristic cost-to-go. The selected node is then moved to the closed list $\mathcal{C}$ and expanded to generate successors. Crucially, the framework detects when a node is revisited via a shorter path from the root, updates its $g(n)$ value accordingly, and moves it back to $\mathcal{O}$.

**Modifications.** To adapt OCL search for generative planning models, we introduce several modifications to the standard framework. First, we introduce *depth-partitioned selection*, maintaining separate open lists per depth level (determined by $g(n)$) to balance exploration across the solution depth and counteract systematic heuristic overestimation. Second, we perform node expansions via *truncated rollouts* using the generative model $\pi_\theta$ rather than single-step transitions. This allows us to rapidly generate sequences of successor nodes, which effectively prunes large or combinatorial action spaces. Within each rollout, we further apply *adaptive expansion*, using the generative model's token confidence to identify critical decision points. These branching points are expanded immediately, broadening exploration when the model is uncertain and narrowing it when the model is confident. Finally, we integrate a learned distributional heuristic model $h_\phi$ to guide node selection.

Altogether, these components modernize OCL graph search by combining even depth-level coverage with heuristic value guidance, while dynamically constraining breadth based on model confidence. We detail each modification below.

## 4.2. Depth-Partitioned Selection

Heuristic models optimized to predict the cost-to-go $h(n)$ from suboptimal training data systematically overestimate the true cost-to-go $h^*(n)$. To observe how this biases selection, we model the resulting bias as multiplicative, $h(n) \approx \alpha h^*(n)$ for some $\alpha > 1$.

Consider a pathological case: two unvisited frontier nodes, $n_1$ and $n_2$, situated on different branches but projected to yield solutions of identical total length $L$ (i.e., $g(n_1) + h^*(n_1) = g(n_2) + h^*(n_2) = L$). Assuming uniform unit action costs, the occured cost $g(n)$ equals the node's depth $d$. Suppose $n_1$ is shallower than $n_2$ ($g(n_1) < g(n_2)$). Under our bias model, the priority function evaluates to:

$$f(n) = g(n) + \alpha(L - g(n)) = \alpha L - (\alpha - 1)\, g(n). \quad (1)$$

Because $f(n)$ is strictly *decreasing* with respect to $g(n)$, the scores satisfy $f(n_2) < f(n_1)$. Consequently, selecting the node with the lowest $f$-score from a global

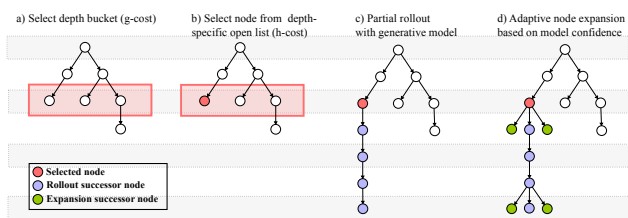

a) Select depth bucket (g-cost)   b) Select node from depth-specific open list (h-cost)   c) Partial rollout with generative model   d) Adaptive node expansion based on model confidence

- Selected node
- Rollout successor node
- Expansion successor node

*Figure 1.* Illustration of one iteration with OCLGen. Note that we maintain a graph structure of states and transitions that is updated after every search iteration.

open list $\mathcal{O}$ (as in standard $A^*$) strictly prefers the deeper node $n_2$, despite both offering paths to equally good solutions. The search thus systematically over-commits to deeper, arbitrary branches, leaving promising shallower alternative routes—where course corrections are most valuable—unexpanded.

To mitigate this, we partition the unvisited nodes by depth, maintaining a separate open list $\mathcal{O}_d$ for each depth level $d$ (where $g(n) = d$). At each iteration, we first select a depth level $d_s$ (Figure 1a), then extract a node $n_s$ from $\mathcal{O}_{d_s}$ based on the heuristic estimate $h_\phi$ (Figure 1b). This two-stage selection ensures even exploration across potential solution lengths. Crucially, at a fixed depth $d$, pairwise comparisons within $\mathcal{O}_d$ depend only on the ranking signal $\alpha h^*(n)$. Thus, our learned heuristic $h_\phi$ provides a useful ranking signal within each depth level despite its absolute overestimation.

We investigate two strategies for selecting $d_s$ from $\{0, 1, \ldots, d_{\max} - 1\}$, where $d_{\max}$ is the depth of the shallowest goal state found so far (or the maximum graph depth if no solution exists): *Uniform selection* samples $d_s$ uniformly at random, while *Scan selection* iterates sequentially from $d = 0$ to $d_{\max} - 1$, ensuring systematic coverage across all levels.

### 4.3. Confidence-based Node Expansion

Having selected a depth level $d_s$ and extracted the highest-scoring node $n_s \in \mathcal{O}_{d_s}$ according to the heuristic $h_\phi$, we perform a truncated rollout starting from $n_s$ with the generative model $\pi_\theta$ (Figure 1c), producing a partial plan segment $(a_0, a_1, \ldots)$. The rollout is capped at a fixed number of tokens, chosen shorter than the longest plan's token sequence in the training data; this reduces computational overhead while still providing useful search guidance. We then decide, action by action, where to branch. For each generated action in the rollout, we compute a confidence value by taking the minimum probability assigned to any of the sampled tokens comprising that action (operator and argument tokens). We then apply *adaptive expansion* (Figure 1d): for each node $n_i$ and outgoing action $a_i$ along the rollout where the action's confidence value falls below a threshold $\tau_{\mathrm{conf}}$, we generate valid successor states as in classical node

expansion and move node $n_i$ to $\mathcal{C}$. Nodes whose outgoing action has confidence above $\tau_{\mathrm{conf}}$ are not expanded, concentrating computational resources on the critical decision points where exploration is most valuable. This contrasts with best-of-N sampling, which redraws complete plans and repeatedly regenerates the high-confidence segments shared across samples. Low confidence flags the decision points worth exploring; one plausible source is that the training data contains multiple alternative solutions to similar problems, leading the model to spread probability across competing actions.

To quantify the resulting savings, assume a constant branching factor $b$. Exhaustive expansion explores $\sum_{k=1}^{d} b^k$ nodes to reach depth $d$. Adaptive expansion instead follows confident actions greedily (local branching factor 1) and branches exhaustively otherwise (local branching factor $b$). If the model is confident at fraction $\gamma \in [0, 1]$ of decision steps, the expected branching factor becomes

$$\tilde{b} = (1 - \gamma)\, b + \gamma \qquad (2)$$

and, assuming branching decisions are approximately independent across nodes, the expected number of explored nodes to depth $d$ becomes $\sum_{k=1}^{d} \tilde{b}^k$. At $\gamma = 0$, Equation (2) recovers full expansion; at $\gamma = 1$, the sum collapses to $d$ (a single path). Because the node count grows as $\tilde{b}^k$, even a modest $\gamma$ compounds into substantial savings with depth, making rollouts with adaptive expansion especially effective in domains with large combinatorial action spaces.

All newly generated but unexpanded nodes are assigned to their respective depth-partitioned open lists $\mathcal{O}_d$ (or the closed list $\mathcal{C}$ if their state has already been visited). Consistent with standard OCL search, we update $g$-costs and reassign nodes to the appropriate depth list $\mathcal{O}_d$.

### 4.4. Distributional Heuristic Model

We frame cost-to-go prediction as a discrete classification task over the set of possible remaining plan lengths $\mathcal{L} = \{0, 1, \ldots, L_{\max}\}$. The model, parameterized by $\phi$, outputs a conditional probability distribution $P_\phi(c \mid s, s_g)$ representing the predicted probability that the true remaining plan length from state $s$ to goal $s_g$ is exactly $c \in \mathcal{L}$. This model is trained via maximum likelihood estimation using a standard cross-entropy loss on $\mathcal{D}_{\mathrm{train}}$. The scalar heuristic value $h_\phi(n)$ used downstream for search is extracted as a summary statistic from this distribution.

Because the model is trained on suboptimal demonstrations, $P_\phi$ concentrates on suboptimal plan lengths, leaving the true optimal cost-to-go $h^*$ in its lower tail. Depth partitioning (Section 4.2) already neutralizes the bias that overestimation induces when comparing nodes across depths; the choice of summary statistic plays a complementary role within a depth. Since search ultimately retains only the best plan

reachable from a node, we want $h_\phi$ to rank nodes by their best attainable completion rather than their typical, suboptimal one. We therefore summarize $P_\phi$ by its lower $k$-th percentile rather than the mode, targeting the optimistic tail. While selecting the $k$-th percentile is not universally more optimistic than the mode for all distribution shapes, it empirically provides a useful ranking signal in our domains (see Appendix F.3).

For unexpanded nodes $n_i$ along a rollout (where confidence exceeded $\tau_{conf}$), we minimize forward passes through the heuristic model by backing up the estimate from the nearest downstream expanded node $n_j$. Specifically, we set $h(n_i) = h_\phi(n_j) + g(n_j) - g(n_i)$, unless $n_i$ is the final state of the rollout.

### 4.5. Base Model Improvements via Action Compilation

Building on Rossetti et al. (2024b), we introduce a new training data augmentation strategy that samples a random offset for each ground-truth plan, computes the corresponding intermediate state using a formal plan validator (Howey et al., 2004), and constructs training examples from these intermediate states paired with their remaining action sequences. This exposes the model to diverse starting states and shorter planning horizons, improving generalization necessary when calling the model on intermediate states within the search graph. We apply this strategy when training both policy and heuristic models. Empirical validation is provided in Appendix F.1.

### 4.6. Algorithm Overview

We structure OCLGEN as an anytime algorithm — one that returns a valid solution at any point and continues refining it as more compute becomes available. Initially, we execute $N_{init}$ full rollouts from $s_0$ to establish baseline solutions and populate the search graph with goal-reaching trajectories. The algorithm then iteratively refines these solutions through depth-partitioned search until a compute budget is reached (e.g., a maximum number of iterations or wall-clock time). Algorithm 1 provides pseudocode for the main search loop, and Figure 1 illustrates each step visually. Each iteration proceeds as follows: (a) `select_depth` chooses a depth level to focus expansion; (b) `select_node` picks a promising node from the corresponding open list; (c) `partial_rollout` generates a truncated trajectory from the selected node toward $s_g$; (d) `adaptive_expansion` uses rollout confidence to decide which intermediate states to expand; and (e) newly generated nodes are scored and added to the graph, with $g$-costs updated and nodes reassigned to their depth-partitioned open lists $\mathcal{O}_d$ or the closed list $\mathcal{C}$ if their state has already been visited. Upon termination, the algorithm returns the lowest-cost plan found.

---

**Algorithm 1** Depth-partitioned OCL for Generative Models

**Input**: Initial state $s_0$, goal $s_g$, iterations $n_{iters}$, initial rollouts $N_{init}$, generative model $\pi_\theta$, heuristic model $h_\phi$

  # Initialize empty search graph
  $G \leftarrow$ `SearchGraph()`
  # Add root node to graph
  $G$.`add(Node`$(s_0, g = 0, h = h_\phi(s_0, s_g))$`)`
  # Execute initial rollouts to populate graph
  $G \leftarrow$ `initial_rollouts`$(G, \pi_\theta, s_0, s_g, N_{init})$
  # Main search loop
  **for** $i_{iter}$ in $1 \ldots n_{iters}$ **do**
    # Select a depth level
    $d_s \leftarrow$ `select_depth`$(G, i_{iter})$     (Fig. 1a)
    # Select node from depth-specific open list
    $n_s \leftarrow$ `select_node`$(G, d_s)$     (Fig. 1b)
    # Generate new nodes from rollout
    $S_{roll} \leftarrow$ `partial_rollout`$(\pi_\theta, n_s, s_g)$   (Fig. 1c)
    # Generate new child nodes by expansion
    $S_{exp} \leftarrow$ `adaptive_expansion`$(n_s, s_g, S_{roll})$   (Fig. 1d)
    # Query $h_\phi$ on expanded nodes; assign heuristics via back-propagation to remaining rollout nodes
    $\mathcal{H} \leftarrow$ `assign_heuristics`$(S_{exp}, S_{roll}, h_\phi, s_g)$
    # Add new nodes to graph
    **for** $(s_i, h_i)$ in $\mathcal{H}$ **do**
      $G$.`add(Node`$(s_i, g = g(\text{parent}(s_i))+1, h = h_i)$`)`
    **end for**
  **end for**
  **return** $G$.`get_best_plan()`

---

## 5. Experiments

We provide a large-scale evaluation of test-time inference methods for generative planning in classical AI planning domains. Our experiments address the following questions:

1. Does OCLGEN produce shorter plans than baseline methods for a given compute budget while maintaining high completion rates?
2. Does OCLGEN significantly increase the proportion of generated solutions that are optimal?
3. Which components of our method contribute most to its effectiveness?
4. To what extent is OCLGEN suitable as the basis for a model self-improvement framework?

### 5.1. Test-time Inference Benchmark

**Planning Domains.** We consider the following domains with varying challenges and levels of complexity:

- **Blocksworld**: A classic domain where blocks must be rearranged into goal configurations. Optimal planning is NP-hard (Slaney & Thiébaux, 2001).
- **Logistics**: A transportation domain where packages are delivered between locations using trucks and airplanes across multiple cities.
- **Labyrinth**: A navigation domain on a grid where the agent can move between connected cells or shift entire rows/columns, with cells wrapping around edges.

- **Sokoban**: A puzzle domain where an agent pushes boxes to target locations while navigating around walls. Solving Sokoban is known to be PSPACE-complete (Culberson, 1998).

**Training and Evaluation Setup.** For each domain, we train $\pi_\theta$ and $h_\phi$ on $10^5$ problem instances with suboptimal solutions generated by Fast Downward (LAMA-first configuration (Richter & Westphal, 2010)). Both models use the data augmentation described in Section 4.5. We evaluate on 1000 held-out test instances per domain. We use the GPT-2 (Radford et al., 2019) architecture to implement $\pi_\theta$, and a smaller BERT encoder (Devlin et al., 2019) for $h_\phi$. Both models are trained using the standard cross-entropy loss. Final models are selected based on the checkpoint with the lowest validation loss. Further details on model architecture and training hyperparameters are provided in App. C.

**Baselines.** We compare against the following baselines:

- **MCTS (full rollouts)**: Monte Carlo Tree Search using $\pi_\theta$ for rollouts and with PUCT selection. We use $h_\phi$ to compute values of unsuccessful rollouts.
- **MCTS (partial rollouts)**: MCTS with shorter rollouts, using $h_\phi$ to obtain a value for backpropagation from the last state of the rollout.
- **OCL-Anytime A\***: Classical A\* search using $h_\phi$ as the heuristic, run in anytime mode.
- **OCL-GBFS**: Greedy best-first search using $h_\phi$ as the heuristic for node prioritization.
- **Best-of-N**: Generates a set of plan sequences with $\pi_\theta$ given a runtime limit and returns the shortest valid one.
- **FD-LAMA-anytime**: Fast Downward with LAMA configuration (Richter & Westphal, 2010) which returns the best solution within the time limit.
- **FD-LAMA-first**: This baseline returns the first solution found by FD-LAMA-anytime. This method generated our data for generative model pretraining.
- **FD-optimal**: Fast Downward with the LM-Cut heuristic (48h timeout), establishing reference sets of optimal plans for each domain.

All learned methods use the same generative and heuristic models, isolating the effect of the inference algorithm. A 10-minute timeout is given per problem (except for FD solvers). For OCLGEN, we evaluate two different depth selection strategies *uniform* and *scan* (Section 4.2). We empirically determined an action confidence threshold $\tau_{\text{conf}}=0.95$ for all domains, except for Logistics where we use $\tau_{\text{conf}}=0.2$. More details on hyperparameter choices for inference are given in Appendix D.

**Plan Lengths.** Table 1 presents completion rates and plan lengths across all domains. OCLGEN achieves the best combination of completion rate and plan quality among all methods operating within the 10-minute budget per problem instance. Both OCLGEN variants attain near-perfect completion rates (99.7–100%) while producing consistently shorter plans than competing approaches. Notably, OCLGEN reduces average plan length by 19.6% compared to MCTS with full rollouts on Blocksworld and by 17.8% on Labyrinth. While MCTS with partial rollouts and search-based methods (OCL-Anyt. A\*, OCL-GBFS) achieve competitive results on simpler domains, they fail to scale to the more challenging Logistics and Sokoban benchmarks, with completion rates dropping to 1.3–37.3%. Best-of-N sampling achieves high completion rates but consistently yields longer plans than OCLGEN across all domains. The FD-optimal solver, even with a 48-hour time limit, solves only 16.9–63% of instances on Blocksworld, Logistics, and Sokoban, underscoring the inherent difficulty of these domains. Finally, the *uniform* and *scan* selection strategies yield comparable performance, while *uniform* achieves slightly shorter plans overall except for Sokoban.

**Solution Optimality.** Table 2 examines plan statistics on the subset of instances for which optimal solutions are known (solved by FD-optimal within 48h). OCLGEN yields the most optimal solutions among all learning methods, solving 83.8% (528/630) of Blocksworld, 61.5% (104/169) of Logistics, 98.8% (988/1000) of Labyrinth, and 77.1% (377/489) of Sokoban with the *uniform* variant. The *scan* variant performs comparably, marginally improving the Sokoban rate to 78.5% (384/489). This represents a $2.9\times$ improvement over MCTS with full rollouts on Blocksworld and a $1.6\times$ improvement on Labyrinth. Notably, OCLGEN's (*uniform*) average plan lengths on solved instances closely approach those of the FD-optimal solver: within 1.7% on Blocksworld (30.64 vs. 30.12), 5.2% on Logistics (28.73 vs. 27.31), 0.2% on Labyrinth (12.99 vs. 12.96), and 3.0% on Sokoban (48.45 vs. 47.03). While OCL-Anyt. A\* achieves competitive optimality rates on Blocksworld (336/630) and Labyrinth (975/1000), it fails to solve the majority of Logistics and Sokoban instances within the time limit. On this subset, FD-LAMA-anytime (10min) yields more optimal solutions in Blocksworld, Logistics, and Sokoban. However, these instances favor lower-complexity problems solvable by FD-optimal within 48 hours. Crucially, on the full test set—which spans higher-complexity problems—OCLGEN scales significantly better, producing shorter plans than FD-LAMA-anytime across all four domains (Table 1). Moreover, we later show that our method can distill its search results back into the model, surpassing FD-LAMA-anytime even on this subset after recursive self-improvement (Section 5.4). Overall, these results demonstrate that OCLGEN significantly increases the percentage of optimal solutions while maintaining high completion rates.

| Method ($t_{max}$=10min) | Blocksworld | | Logistics | | Labyrinth | | Sokoban | |
|---|---|---|---|---|---|---|---|---|
| | Comp.[%] | Length | Comp.[%] | Length | Comp.[%] | Length | Comp.[%] | Length |
| **OCLGen** (uniform) | 100.0 | 43.88 (± 0.68) | 100.0 | 155.83 (± 3.57) | 100.0 | 12.99 (± 0.13) | 99.9 | 128.67 (± 3.15) |
| **OCLGen** (scan) | 100.0 | 44.10 (± 0.70) | 100.0 | 157.63 (± 3.51) | 100.0 | 13.00 (± 0.13) | 99.7 | 128.10 (± 3.15) |
| MCTS (full rollouts) | 100.0 | 54.56 (± 0.93) | 100.0 | 158.75 (± 3.52) | 100.0 | 15.81 (± 0.23) | 99.8 | 131.30 (± 3.23) |
| MCTS (partial rollouts) | 100.0 | 53.81 (± 0.91) | 29.3 | 40.31 (± 1.30) | 100.0 | 14.77 (± 0.19) | 37.3 | 38.34 (± 0.93) |
| OCL-Anyt. A* | 71.1 | 37.07 (± 0.71) | 1.6 | 2.81 (± 0.70) | 100.0 | 13.04 (± 0.13) | 33.5 | 37.30 (± 1.10) |
| OCL-GBFS | 89.5 | 65.23 (± 1.50) | 1.3 | 2.46 (± 0.85) | 99.2 | 18.03 (± 0.43) | 35.9 | 48.38 (± 1.71) |
| Best-of-N | 99.9 | 61.56 (± 1.03) | 100.0 | 157.00 (± 3.47) | 100.0 | 17.42 (± 0.25) | 98.8 | 132.25 (± 3.22) |
| FD-LAMA-anytime ($t_{max}$=10min) | 100.0 | 45.05 (± 0.73) | 100.0 | 161.00 (± 3.57) | 100.0 | 19.77 (± 0.41) | 99.4 | 141.43 (± 3.84) |
| FD-LAMA-anytime ($t_{max}$=20min) | 100.0 | 44.85 (± 0.72) | 100.0 | 160.93 (± 3.57) | 100.0 | 16.25 (± 0.32) | 100.0 | 143.43 (± 3.84) |
| FD-LAMA-first ($t_{max}$=20min) | 100.0 | 79.43 (± 1.50) | 100.0 | 165.32 (± 3.42) | 100.0 | 25.78 (± 0.50) | 100.0 | 149.05 (± 3.62) |
| FD-optimal ($t_{max}$=48h) | 63.0 | 30.12 (± 0.48) | 16.9 | 27.31 (± 1.48) | 100.0 | 12.96 (± 0.12) | 48.9 | 47.03 (± 1.08) |

*Table 1.* Benchmark on unseen problems (1000 per domain). Comp.: Completion (%); Length: Mean plan length (± std. error).

| Method ($t_{max}$=10min) | Blocksworld | | Logistics | | Labyrinth | | Sokoban | |
|---|---|---|---|---|---|---|---|---|
| | Optimal | Length (solved) | Optimal | Length (solved) | Optimal | Length (solved) | Optimal | Length (solved) |
| **OCLGen** (uniform) | 528 / 630 | 30.64 (630 / 630) | 104 / 169 | 28.73 (169 / 169) | 988 / 1000 | 12.99 (1000 / 1000) | 377 / 489 | 48.45 (489 / 489) |
| **OCLGen** (scan) | 522 / 630 | 30.67 (630 / 630) | 99 / 169 | 28.99 (169 / 169) | 987 / 1000 | 13.00 (1000 / 1000) | 384 / 489 | 48.29 (489 / 489) |
| MCTS (full rollouts) | 180 / 630 | 36.48 (630 / 630) | 64 / 169 | 29.70 (169 / 169) | 605 / 1000 | 15.81 (1000 / 1000) | 291 / 489 | 49.52 (488 / 489) |
| MCTS (partial rollouts) | 186 / 630 | 36.48 (630 / 630) | 64 / 169 | 26.23 (160 / 169) | 671 / 1000 | 14.77 (1000 / 1000) | 271 / 489 | 38.34 (373 / 489) |
| OCL-Anyt. A* | 336 / 630 | 31.25 (594 / 630) | 16 / 169 | 2.81 (16 / 169) | 975 / 1000 | 13.04 (1000 / 1000) | 293 / 489 | 36.55 (333 / 489) |
| OCL-GBFS | 146 / 630 | 42.69 (626 / 630) | 12 / 169 | 3.46 (13 / 169) | 589 / 1000 | 18.03 (992 / 1000) | 160 / 489 | 43.85 (341 / 489) |
| Best-of-N | 85 / 630 | 42.40 (630 / 630) | 59 / 169 | 29.93 (169 / 169) | 444 / 1000 | 17.42 (1000 / 1000) | 251 / 489 | 50.87 (484 / 489) |
| FD-LAMA-anytime ($t_{max}$=10min) | 591 / 630 | 30.53 (630 / 630) | 105 / 169 | 28.56 (169 / 169) | 459 / 1000 | 19.77 (1000 / 1000) | 421 / 489 | 47.84 (489 / 489) |
| FD-LAMA-anytime ($t_{max}$=20min) | 601 / 630 | 30.48 (630 / 630) | 106 / 169 | 28.51 (169 / 169) | 653 / 1000 | 16.25 (1000 / 1000) | 441 / 489 | 47.58 (489 / 489) |
| FD-LAMA-first ($t_{max}$=20min) | 62 / 630 | 52.45 (630 / 630) | 47 / 169 | 31.21 (169 / 169) | 249 / 1000 | 25.78 (1000 / 1000) | 121 / 489 | 58.08 (489 / 489) |
| FD-optimal ($t_{max}$=48h) | 630 / 630 | 30.12 (630 / 630) | 169 / 169 | 27.31 (169 / 169) | 1000 / 1000 | 12.96 (1000 / 1000) | 489 / 489 | 47.03 (489 / 489) |

*Table 2.* Benchmark on subset of known optimal problems. Optimal: num. solved optimally; Length: Plan length (num. solved).

**Plan Quality Convergence.** Figure 2 shows the average plan length and completion rates against runtime. Note that average plan lengths initially increase as shorter problems tend to be solved first. Across all domains, OCLGEN achieves shorter plans significantly faster than baseline methods. On Blocksworld, it reaches an average plan length of 50 steps within 30 seconds, a quality that MCTS and Best-of-N fail to match even after 5 minutes of runtime. Similarly, in Labyrinth, OCLGEN converges to near-optimal lengths (∼13 steps) in under 200 seconds, while MCTS requires the full 600-second budget to plateau at a substantially higher length. This pattern holds in Logistics and Sokoban, where OCLGEN consistently achieves lower plan lengths after completion rates stabilized. The steeper descent of OCLGEN's curves indicates more effective use of compute: each additional second of runtime yields greater plan quality improvements compared to the baselines. Importantly, as shown in Figure 3, this improved plan quality does not come at the cost of completion rate. OCLGEN reaches near-perfect completion rates within seconds on Blocksworld, Labyrinth, and Logistics, and remains competitive with MCTS on Sokoban.

## 5.2. Generalization Beyond Training Data

Table 3 evaluates OCLGEN (uniform) on problem instances where the training data planner (FD-LAMA with a 20-minute budget) failed to find a solution. OCLGEN solves

| Method ($t_{max}$=10min) | Logistics | | Sokoban | |
|---|---|---|---|---|
| | Comp.[%] | Length | Comp.[%] | Length |
| **OCLGen** (uniform) | 100.0 | 374.85 (± 4.71) | 93.6 | 348.94 (± 3.66) |
| Best-of-N [$N_{max}$=∞] | 98.5 | 376.09 (± 3.59) | 90.8 | 364.27 (± 4.19) |

*Table 3.* Completion rate and plan length statistics of OCLGEN (uniform) and Best-of-N on unseen problems for which the training data generation with FD-LAMA-first ($t_{max}$=20min) failed (number of problems: 66 for Logistics and 250 for Sokoban). Comp.: Completion (%); Length: Mean plan length (± std. error).

100% of these challenging Logistics instances and 93.6% of Sokoban instances. This indicates that the generative model learns transferable strategies rather than merely overfitting the training data. Furthermore, OCLGEN consistently outperforms Best-of-N sampling on these difficult instances, improving completion rates by 1.5 percentage points on Logistics and 2.8 percentage points on Sokoban, while also producing shorter plans (374.85 vs. 376.09 on Logistics; 348.94 vs. 364.27 on Sokoban).

## 5.3. Ablations

To evaluate the contribution of each component in OCLGEN, we systematically ablate three key design choices across all four domains: (1) *depth partitioning*, (2) *adaptive expansion*, and (3) the *percentile-based cost-to-go estimator*. For each ablation, we remove or replace a single component while keeping all other settings fixed, us-

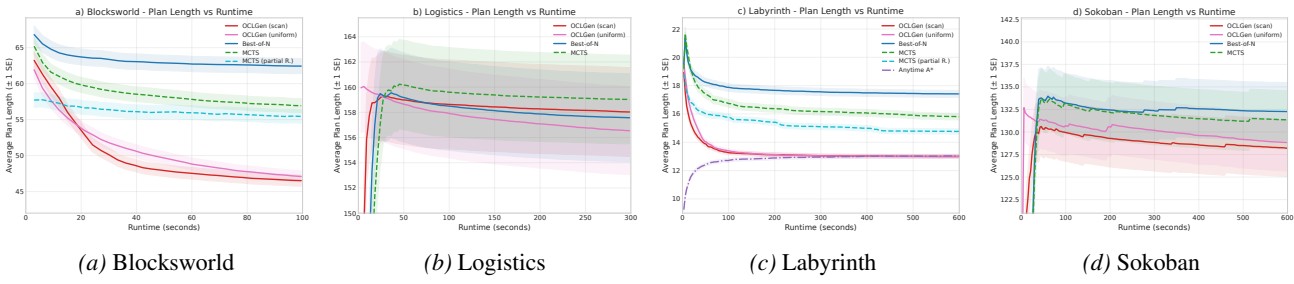

*Figure 2.* Plan length over time across all domains. OCLGEN rapidly converges to shorter plans compared to baseline methods.

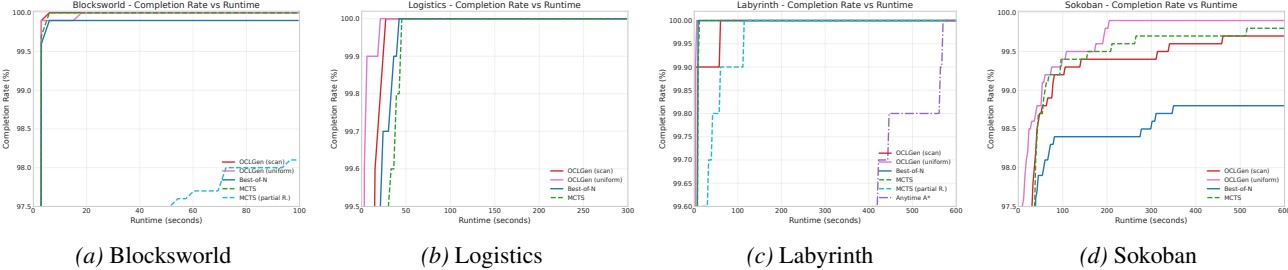

*Figure 3.* Completion rate over time across all domains.

| Domain | Metric | w/o Depth selection | w/o Adaptive expansion | w/o Percentile estimate (mode) |
|---|---|---|---|---|
| **Blocksworld** | Optim. | 366 / 630 | 496 / 630 | 486 / 630 |
| | Comp.[%] | 100.0 | 100.0 | 100.0 |
| | Length | 48.81 (± 0.87) | 44.35 (± 0.69) | 44.68 (± 0.70) |
| **Logistics** | Optim. | 78 / 169 | 98 / 169 | 98 / 169 |
| | Comp.[%] | 99.9 | 100.0 | 100.0 |
| | Length | 159.23 (± 3.52) | 155.82 (± 3.47) | 156.11 (± 3.48) |
| **Labyrinth** | Optim. | 979 / 1000 | 983 / 1000 | 980 / 1000 |
| | Comp.[%] | 100.0 | 100.0 | 100.0 |
| | Length | 13.08 (± 0.15) | 13.02 (± 0.13) | 13.03 (± 0.13) |
| **Sokoban** | Optim. | 353 / 489 | 341 / 489 | 350 / 489 |
| | Comp.[%] | 99.7 | 100.0 | 99.8 |
| | Length | 129.39 (± 3.17) | 129.73 (± 3.15) | 129.52 (± 3.17) |

*Table 4.* Ablation study across problem domains. Comp.: Completion rate (%); Length: Plan length (mean ± std. error).

ing OCLGEN (uniform) as the base configuration. Table 4 reports completion rates and plan lengths across 1000 test instances per domain, again given a 10-min runtime limit. Each component provides incremental improvements that collectively drive the full method's performance.

**Impact of Depth Partitioning.** Table 4 shows that removing depth-based selection leads to notably longer plans, particularly on Blocksworld (48.81 vs. 43.88) and Logistics (159.23 vs. 155.83). Without depth partitioning, the search tends to over-exploit deep branches, reducing the diversity of explored trajectories and ultimately yielding suboptimal solutions.

**Confidence-based Adaptive Expansion.** The adaptive expansion mechanism dynamically decides when to expand

the search tree based on rollout confidence. Removing this component yields comparable completion rates and overall longer plans (e.g. 44.35 vs. 43.88 on Blocksworld). The adaptive threshold allows OCLGEN to allocate more computational effort to uncertain regions of the search space while quickly committing to high-confidence trajectories, leading to an improvement in plan quality.

**Influence of Heuristic Point Estimator.** We compare our percentile-based cost-to-go estimate against using the mode of the predicted distribution. Table 4 shows that replacing the percentile estimate with the mode results in marginally longer plans across all domains. While the differences are modest, the percentile-based estimator yields more optimistic cost-to-go estimates that counteract overestimation bias, improving plan quality on Blocksworld (43.88 vs. 44.68) and Logistics (155.83 vs. 156.11).

### 5.4. OCLGen for Model Self-Improvement

We evaluate the use of OCLGEN for recursive model self-improvement. Given a random subset of training problems, we run our search algorithm for a fixed runtime budget on each problem instance to compute improved plans, then fine-tune both $\pi_\theta$ and $h_\phi$ on this data. This procedure is applied recursively, using updated model weights from the previous iteration to generate improved plans on newly sampled problems. We compare OCLGEN (uniform) and MCTS (full rollouts) for $n_{\text{loop}}=3$ iterations of self-improvement in Blocksworld and Sokoban. At each iteration, we sample 3000 problems and run the search method on each of these given a time limit of 3 minutes in Blocksworld and 5

|  | $i_{\text{loop}}$ | OCLGen (uniform) Comp.[%] | Length | MCTS (full rollouts) Comp.[%] | Length |
|---|---|---|---|---|---|
| **Blocksw.** | 0 | 99.9 | 69.51 (± 1.24) | 99.9 | 69.51 (± 1.24) |
|  | 1 | 99.2 | 47.70 (± 0.84) | 99.9 | 62.20 (± 1.18) |
|  | 2 | 99.6 | 41.81 (± 0.58) | 99.9 | 57.30 (± 1.06) |
|  | 3 | 99.3 | 41.40 (± 0.57) | 99.9 | 53.15 (± 0.98) |
| **Sokob.** | 0 | 95.8 | 135.00 (± 3.38) | 95.8 | 135.00 (± 3.38) |
|  | 1 | 95.2 | 128.41 (± 3.25) | 95.1 | 128.89 (± 3.24) |
|  | 2 | 93.5 | 123.93 (± 3.17) | 93.9 | 125.94 (± 3.18) |
|  | 3 | 93.4 | 121.88 (± 3.11) | 93.9 | 124.48 (± 3.16) |

*Table 5.* Self-improvement plan statistics on the test dataset (1000 samples) using best-of-N sampling ($N$=10). Comp.: Completion rate (%); Length: Plan length (mean ± std. error).

|  |  | OCLGen (uniform) | MCTS (full rollouts) |
|---|---|---|---|
| **Blocksw.** | Plan Length | 40.74 (± 0.55) | 44.89 (± 0.70) |
|  | Compl. [%] | 100.0 | 100.0 |
|  | Plan length (optimal) | 30.12 (± 0.48) | 31.14 (± 0.53) |
|  | Optimal | 630 / 630 (100.0%) | 450 / 630 (71.4%) |
| **Sokob.** | Plan Length | 123.60 (± 3.02) | 125.06 (± 3.08) |
|  | Compl. [%] | 99.8 | 99.3 |
|  | Plan length (optimal) | 47.17 (± 1.09) | 47.55 (± 1.10) |
|  | Optimal | 463 / 489 (94.7%) | 400 / 489 (81.8%) |

*Table 6.* Results on Blocksworld and Sokoban with OCLGEN and MCTS ($t_{\text{max}}$=10min) after $n_{\text{loop}} = 3$ iteration of model self-improvement on the test sets from Table 1 (1000 samples). Comp.: Completion rate (%); Length: Plan length (mean ± std. error). Optimal: number solved optimally.

minutes in Sokoban (further details in Appendix E).

Table 5 reports the test set evaluation of the finetuned generative models after each self-improvement iteration using Best-of-N sampling ($N$=10). Both methods maintain near-full completion on Blocksworld, but OCLGEN consistently produces shorter plans, providing a higher quality training signal. On Sokoban, a similar pattern is observed, although completion rates drop for both methods, presumably because finetuning on improved plans reduces policy entropy, decreasing diversity within the Best-of-N batch.

Table 6 details the test set evaluation of our final models, which utilize test-time search after three rounds of self-improvement. On Blocksworld, OCLGEN yields significantly shorter plans (40.74 vs. 44.89) and achieves 100% optimal solutions (vs. 71.4% for MCTS). On Sokoban, our method produces shorter plans (123.60 vs. 125.06) and achieves 94.7% optimal solutions (vs. 81.8% for MCTS), while recovering from the completion rate drop observed with the model alone to achieve 99.8% completion. Notably, on the problem subsets with known optimal solutions, self-improvement with OCLGEN surpasses the FD-LAMA-anytime solver (601/630 on Blocksworld and 441/489 on Sokoban at a 20-minute budget; Table 2). These results demonstrate the suitability of OCLGEN for efficient model self-improvement.

|  | Blocksworld | | Sokoban | |
|---|---|---|---|---|
| $n_{\text{loop}}$ | 0 | 3 | 0 | 3 |
| Mode | 8.68 (± 0.24) | 0.44 (± 0.02) | 5.28 (± 0.17) | 3.05 (± 0.11) |
| Perc. | 4.07 (± 0.12) | 0.66 (± 0.02) | 3.31 (± 0.12) | 2.51 (± 0.08) |

*Table 7.* MAE (mean absolute error) ± std. error of heuristic models on the subset of known optimal solutions. Values are reported for both mode-based and percentile-based heuristics.

### 5.5. Accuracy of Heuristic Model

Table 7 reports mean absolute error (MAE) statistics for Blocksworld and Sokoban before and after self-improvement. The MAE is computed on the subset of instances with known optimal solutions, using either the *mode* or *k-th percentile* to derive scalar values from the predicted cost-to-go distributions ($k = 3$ and $k = 10$ for Blocksworld and Sokoban, respectively). As expected, percentile-based estimates yield lower MAE than mode-based estimates for all base models ($n_{\text{loop}} = 0$), reducing the gap to optimality. This behavior holds consistently across all four domains (see Table 15 in the Appendix).

Self-improvement via OCLGEN ($n_{\text{loop}} = 3$) substantially reduces the model's MAE – for example, from 4.07 to 0.66 on Blocksworld and from 3.31 to 2.51 on Sokoban (percentile method). This confirms that OCLGEN not only discovers better solutions across self-improvement rounds, but also successfully refines the heuristic model.

## 6. Conclusion

We present OCLGEN, a test-time search algorithm adapting classical OCL search to autoregressive generative planning models. Key innovations include depth-partitioned selection, partial rollouts with adaptive expansion, and learned distributional heuristics. Across four domains, OCLGEN delivers near-optimal plans at high completion rates, outperforming neurosymbolic and classical baselines on the full test set. Finally, on the known-optimal subsets, recursive self-improvement yields a 100% optimality rate on Blocksworld and 94.7% on Sokoban.

**Future Work.** Our depth selection strategies are currently uninformed, following fixed schedules. Adaptive strategies that concentrate search effort on the most promising depths may yield further gains. Characterizing when recursive self-improvement provably converges to optimal solutions remains an open theoretical question. Scaling to larger problems and generalizing to object counts or grid sizes outside the training distribution remain open challenges, as does transfer to new domains. Finally, our method assumes suboptimal initial solutions for pretraining. Extending OCLGEN to support efficient policy improvement without prior data represents a compelling future direction.

## Impact Statement

This paper presents work whose goal is to advance the field of Machine Learning. There are many potential societal consequences of our work, none which we feel must be specifically highlighted here.

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

# A. Datasets

We construct dedicated datasets for each of the four planning domains evaluated in this work. Each dataset comprises 100k training instances, 1k validation instances for model selection, and 1k held-out test instances. All problem instances are unique and paired with reference solutions obtained using Fast Downward (Helmert, 2003) in the LAMA-first configuration with a 20-minute timeout. Instances that could not be solved within this time limit were excluded from the training datasets.

**Blocksworld**   We generate Blocksworld instances containing 3 to 25 blocks using the generator from (Seipp et al., 2022). All generated instances were solvable by FD-LAMA within the 20-minute budget. The distribution of instances follows a logarithmic scaling with respect to the number of blocks, as the limited number of unique configurations for small block counts precludes a uniform distribution.

**Logistics**   Our Logistics dataset covers a broad range of problem sizes: 1–50 cities with 1–5 locations each, 1–50 packages, 1–10 airplanes, and one truck per city. Instances are sampled uniformly over all valid combinations of these parameters. Fast Downward solved 99.97% of generated instances within the time limit; additional instances were generated to ensure the target dataset sizes.

**Labyrinth**   Labyrinth was introduced in IPC 2023 (Taitler et al., 2024). We generate a plan dataset with $3 \times 3$ and $4 \times 4$ grids, which remain challenging while ensuring sufficient training coverage using FD-LAMA. Problem instances were generated using the official IPC 2023 generator (Eifler & Fišer, 2023).

**Sokoban**   The Sokoban PDDL domain appeared in IPC 2008 and 2011. We constrain our instances to grid sizes between $5 \times 5$ and $14 \times 14$, with 1–10 boxes and up to 10 walls. We use the generator from (Seipp et al., 2022).

## B. Baselines

**Monte Carlo Tree Search.** We implemented Monte Carlo Tree Search using a PUCT-style selection policy similar to Silver et al. (2016). Specifically, we maintain separate Q-values: $Q_{\text{opt}}$ for expected plan length (normalized by plan lengths of neighboring actions) following a transition, and $Q_{\text{sat}}$ for the expected probability of finding a solution. The overall Q-value is computed as a convex combination of $Q_{\text{opt}}$ and $Q_{\text{sat}}$ using a mixing coefficient $\alpha$. To compute the policy prior $P(s, a)$, we determine action probabilities from token sequences by computing the geometric mean of token probabilities for individual action sequences, then normalize over the set of possible actions at a state. The selection policy is $a^* = \arg\max_{a \in \mathcal{A}} \left[ \alpha\, Q_{\text{sat}}(s, a) + (1 - \alpha)\, Q_{\text{opt}}(s, a) + c_{\text{PUCT}}\, P(s, a) \frac{\sqrt{\sum_b N(s,b)}}{1 + N(s,a)} \right]$. For all experiments, we set $\alpha = 0.1$ and $c_{\text{PUCT}} = 1.0$. We also incorporate progressive widening (Coulom, 2007), which significantly improved performance. For MCTS with partial rollouts, we use a maximum token limit of 50. We use the heuristic model $h_\phi$ to estimate the cost-to-go of unsuccessful rollouts.

**Anytime A\*.** We implement a version of Anytime A\* using our learned heuristic $h_\phi$ to guide the search. Similar to our method, we use the lower percentile to obtain heuristic point estimates from distributions. At every step, we select the node with the lowest $f$ value from the open list, where $f = g + h$. The search does not stop once a goal is found but continues until the runtime limit is exhausted or no unvisited nodes remain in the open list.

**Greedy Best-First Search.** This method is similar to Anytime A\* but purely uses the learned heuristic $h_\phi$ to guide the search (i.e., $f = h$). Again, the search continues after finding a goal until the runtime limit is exhausted or no unvisited nodes remain.

**Best-of-N.** This method involves repeated sampling from the generative model $\pi_\theta$. For all experiments, we use a softmax temperature of $T_{\text{softmax}} = 1$ and sample a batch of 10 plans. The generated plans are then validated for syntactic and semantic correctness, and we report the shortest valid plan in the batch. For the experiments, we terminate once the time limit of $t_{\text{max}} = 10$ minutes is reached. Table 8 presents the number of candidate plans per domain generated by Best-of-N within this runtime budget.

| Domain | Number of Generated Plans ($t_{\text{max}}$=10min) |
| --- | --- |
| Blocksworld | 24768.9 ± 11816.0 |
| Logistics | 7243.4 ± 14927.1 |
| Labyrinth | 10953.1 ± 16137.1 |
| Sokoban | 2186.5 ± 4266.4 |

*Table 8.* Number of generated plans (mean ± std. dev.) with Best-of-N for the experiments in Table 1

## C. Training Details

We train domain-specific policy and heuristic cost models on 4 compute instances, each equipped with 8 NVIDIA A100 GPUs (40GB VRAM). Table 9 shows the transformer architecture and training hyperparameters for the generative model. Table 10 shows corresponding hyperparameters for the heuristic cost model. We use the same model architectures across all domains.

| Hyperparameter | Value |
|---|:---:|
| *Architecture* | |
| Type | Transformer Decoder (GPT2) |
| Transformer blocks ($n_{\text{layer}}$) | 12 |
| Attention heads ($n_{\text{head}}$) | 12 |
| Embedding dimension ($n_{\text{embd}}$) | 768 |
| Feedforward dimension ($n_{\text{inner}}$) | 3072 |
| *Optimization* | |
| Learning rate | 5e-5 |
| Scheduler | Cosine |
| Warmup steps | 1000 |
| Optimizer | AdamW |
| Batch size per GPU (BW / Log / Lab / Sok) | 6 / 4 / 6 / 1 |
| *Training* | |
| Max sequence length | 14000 |
| Epochs (BW / Log / Lab / Sok) | 100 / 100 / 100 / 100 |

*Table 9.* Training hyperparameters for the generative model. Unless otherwise indicated, values are identical across domains (BW: Blocksworld, Log: Logistics, Lab: Labyrinth, Sok: Sokoban).

| Hyperparameter | Value |
|---|:---:|
| *Architecture* | |
| Type | Transformer Encoder (BERT) |
| Transformer blocks ($n_{\text{layer}}$) | 6 |
| Attention heads ($n_{\text{head}}$) | 12 |
| Embedding dimension ($n_{\text{embd}}$) | 768 |
| Feedforward dimension ($n_{\text{inner}}$) | 1536 |
| *Optimization* | |
| Learning rate | 5e-5 |
| Scheduler | No |
| Warmup steps | 1000 |
| Optimizer | AdamW |
| Batch size per GPU | 16 |
| *Training* | |
| Max sequence length | 14000 |
| Epochs | 500 |

*Table 10.* Training hyperparameters for the heuristic model. Unless otherwise indicated, values are identical across domains (BW: Blocksworld, Log: Logistics, Lab: Labyrinth, Sok: Sokoban).

## D. Test-time Inference Hyperparameters

For all methods that employ rollouts, we use softmax temperature $T_{\text{softmax}} = 1$ and sample plans in batches of 10. To further reduce the branching factor for highly complex problem instances, we use a simple heuristic based on the number of grounded actions in a problem instance. Specifically, if the number of grounded actions exceeds a predefined threshold $\tau_{|\mathcal{A}|}$ (in practice, this occurs only for a small portion of problems in Logistics), we perform expansions in OCLGEN by sampling a batch of actions from $\pi_\theta$ with batch size 32; otherwise, we enumerate all successor states using the grounded operator models.

| Hyperparameter | Value |
| --- | --- |
| $N_{\text{init}}$ | 3 |
| Max tokens per rollout (BW/Log/Lab/Sok) | 50 / 2000 / 50 / 2000 |
| Confidence threshold $\tau_{\text{conf}}$ (BW/Log/Lab/Sok) | 0.95 / 0.2 / 0.95 / 0.95 |
| Heuristic Percentile $k$ (BW/Log/Lab/Sok) | 3 / 10 / 3 / 10 |
| Policy expansion threshold $\tau_{|\mathcal{A}|}$ | $10^7$ |

*Table 11.* Test-time hyperparameters for our method. Unless otherwise indicated, values are identical across domains (BW: Blocksworld, Log: Logistics, Lab: Labyrinth, Sok: Sokoban).

# E. Self-improvement Details

Our self-improvement pipeline runs for $n_{\text{loop}} = 3$ iterations on Blocksworld and Sokoban. Every iteration samples 3000 problems per domain, executing the search method with $t_{\text{max}} = 3$ minutes (Blocksworld) and $t_{\text{max}} = 5$ minutes (Sokoban). We collect the successfully optimized plans to finetune both the generative model and the heuristic using standard cross-entropy loss. To maintain a consistent training set size, instances unsolved within the time limit default to their original suboptimal training plans. The respective training configurations are detailed in Tables 12 and 13.

| Hyperparameter | Value |
|---|---|
| *Optimization* | |
| Learning rate | 1e-5 |
| Optimizer | AdamW |
| Batch size per GPU (BW / Sok) | 6 / 1 |
| *Training* | |
| Epochs (BW/ Sok) | 100 / 50 |
| Finetuning dataset size | 3000 |

*Table 12.* Finetuning hyperparameters for the generative model. Unless otherwise indicated, values are identical across domains.

| Hyperparameter | Value |
|---|---|
| *Optimization* | |
| Learning rate | 1e-5 |
| Optimizer | AdamW |
| Batch size per GPU | 16 / 16 |
| *Training* | |
| Epochs (BW/ Sok) | 100 / 50 |
| Finetuning dataset size | 3000 |

*Table 13.* Finetuning hyperparameters for the heuristic model. Unless otherwise indicated, values are identical across domains.

# F. Additional Results

## F.1. Data Augmentation via Action Compilation

In Section 4.5, we introduce a data augmentation method for training our generative planning model. Given the plan sequences in $\mathcal{D}_{\text{train}}$, we use the operator effects specified in the PDDL domain definition to compile intermediate states along each plan, generating new state-plan pairs. We apply this procedure when training both the generative model and the heuristic model. During batch generation, we first sample a random offset within the plan length, then compute the corresponding intermediate state by sequentially applying operator effects up to this offset. A new training sample is formed by pairing this intermediate state with the remaining plan suffix.

The results in Table 14 compare models trained with and without this state compilation augmentation. The evaluation, performed on 1000 unseen problem instances per domain, uses Best-of-N sampling with $N = 10$ for both model variants. Outside of Blocksworld, where the two variants perform comparably, augmentation improves completion rate on Logistics, Labyrinth, and Sokoban, and shortens plans on Labyrinth and Sokoban. Overall, exposing the model to intermediate states during training enhances its ability to generalize across planning instances.

| | Blocksworld | | Logistics | | Labyrinth | | Sokoban | |
|---|---|---|---|---|---|---|---|---|
| GM Training Treatment | Comp.[%] | Length | Comp. [%] | Length | Comp. [%] | Length | Comp. [%] | Length |
| w/o compilation | 100.0 | 69.06 ($\pm$ 1.25) | 99.0 | 161.83 ($\pm$ 3.60) | 98.60 | 25.44 ($\pm$ 0.55) | 91.2 | 141.02 ($\pm$ 3.58) |
| w/ compilation | 99.9 | 69.51 ($\pm$ 1.24) | 99.8 | 162.17 ($\pm$ 3.58) | 99.90 | 24.23 ($\pm$ 0.55) | 95.8 | 135.00 ($\pm$ 3.38) |

*Table 14.* Performance comparison across problem domains for a generative planning model with and without action compilation. Comp.: Completion (%); Length: Plan length (mean $\pm$ std error).

## F.2. Accuracy of the learned cost model

Figures 4–7 show the per-domain distribution of absolute errors between $h_\phi$'s predictions and the held-out (suboptimal) test labels, characterizing how closely the heuristic fits the data-generating planner. Accuracy against the *true optimal* cost-to-go is reported separately in Table 15.

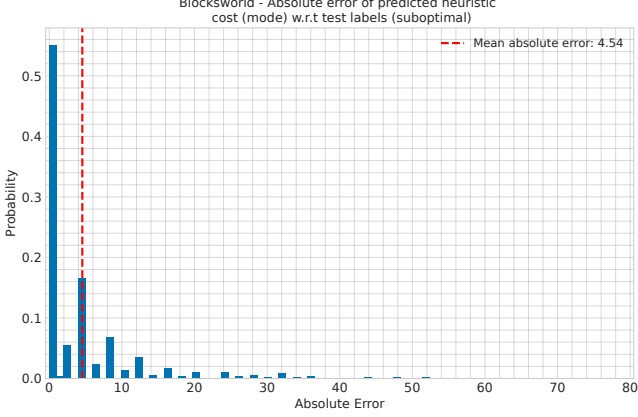

*Figure 4.* Blocksworld - Distribution of absolute errors of learned heuristic cost model with respect to test labels (suboptimal data).

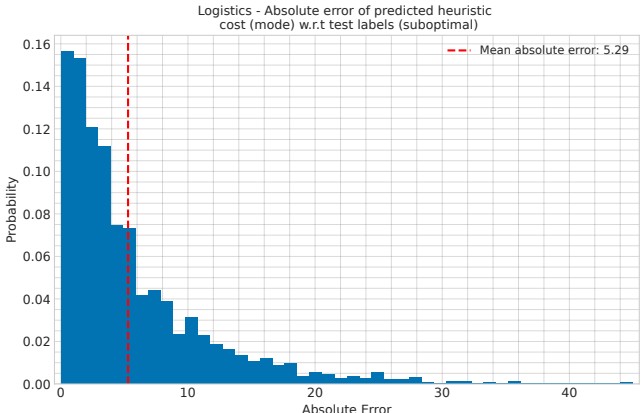

*Figure 5.* Logistics - Distribution of absolute errors of learned heuristic cost model with respect to test labels (suboptimal data).

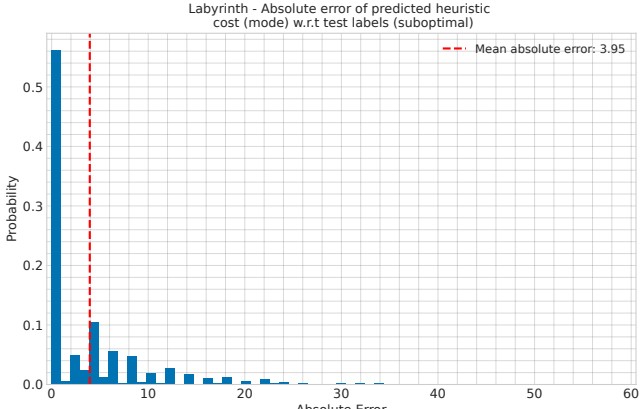

*Figure 6.* Labyrinth - Distribution of absolute errors of learned heuristic cost model with respect to test labels (suboptimal data).

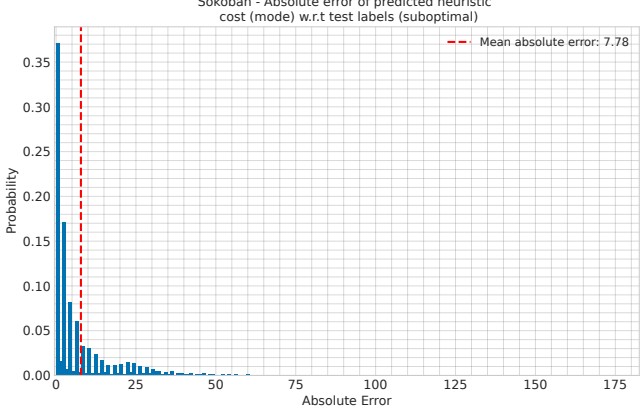

*Figure 7.* Sokoban - Distribution of absolute errors of learned heuristic cost model with respect to test labels (suboptimal data).

## F.3. Illustrations of Learned Cost Distribution

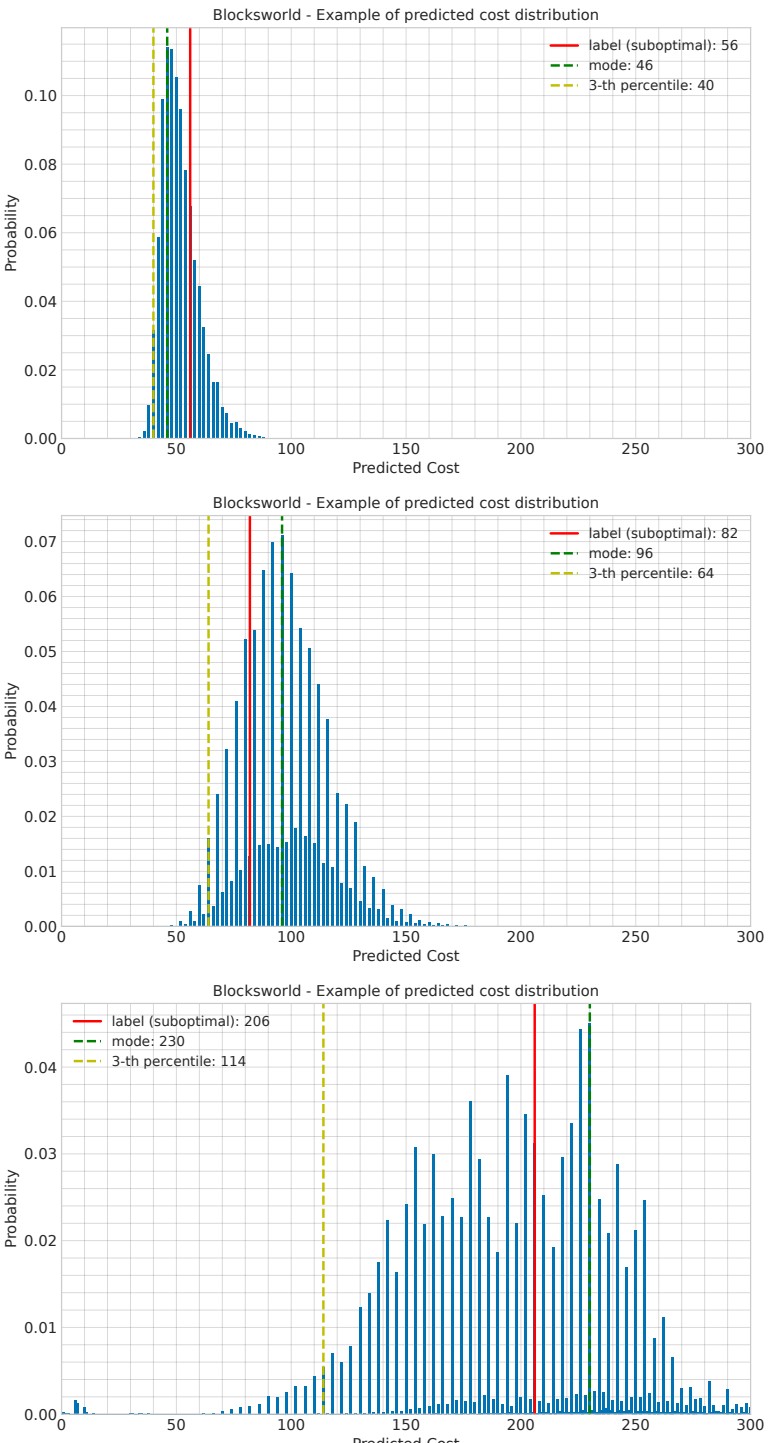

*Figure 8.* Blocksworld - Examples of cost distribution generated by the learned heuristic model.

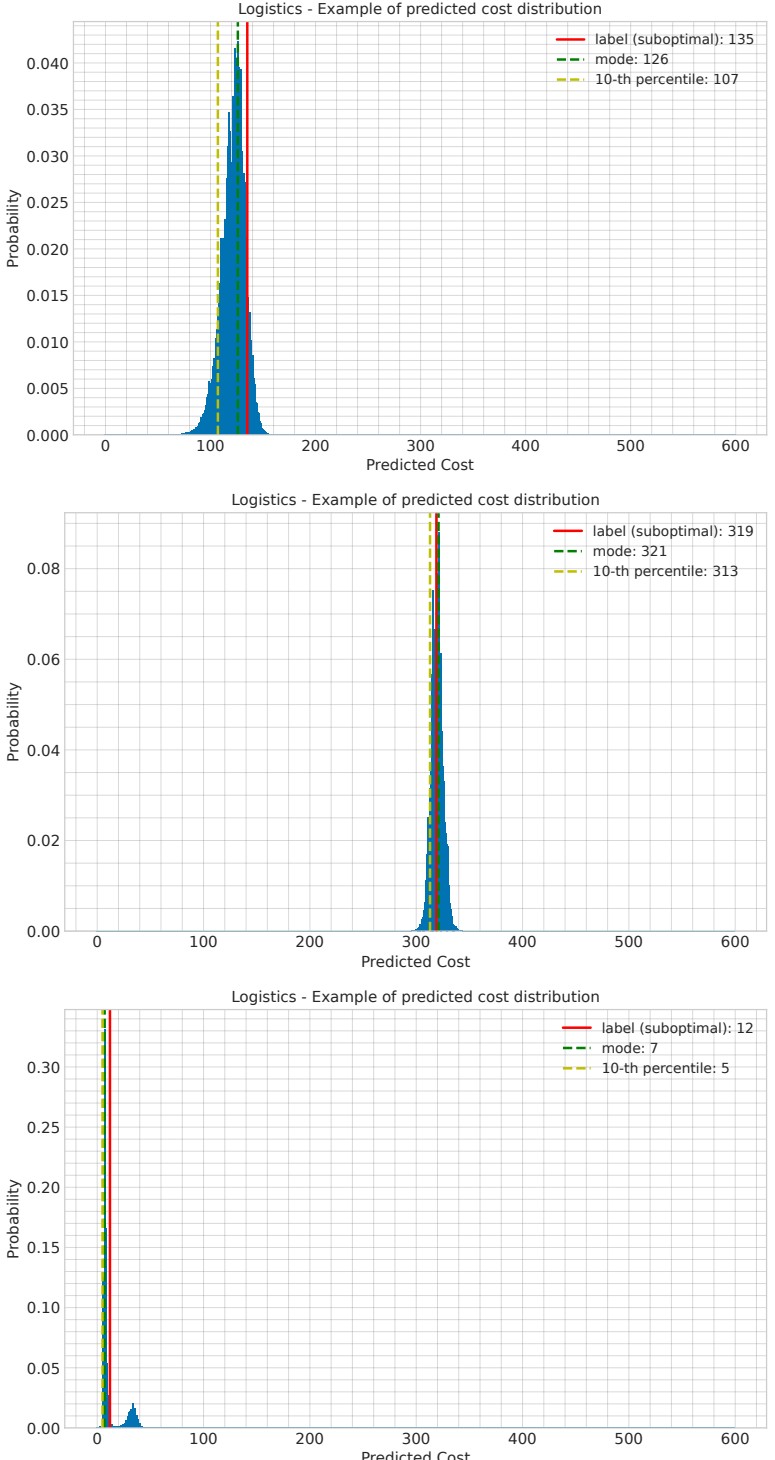

*Figure 9.* Logistics - Examples of cost distribution generated by the learned heuristic model.

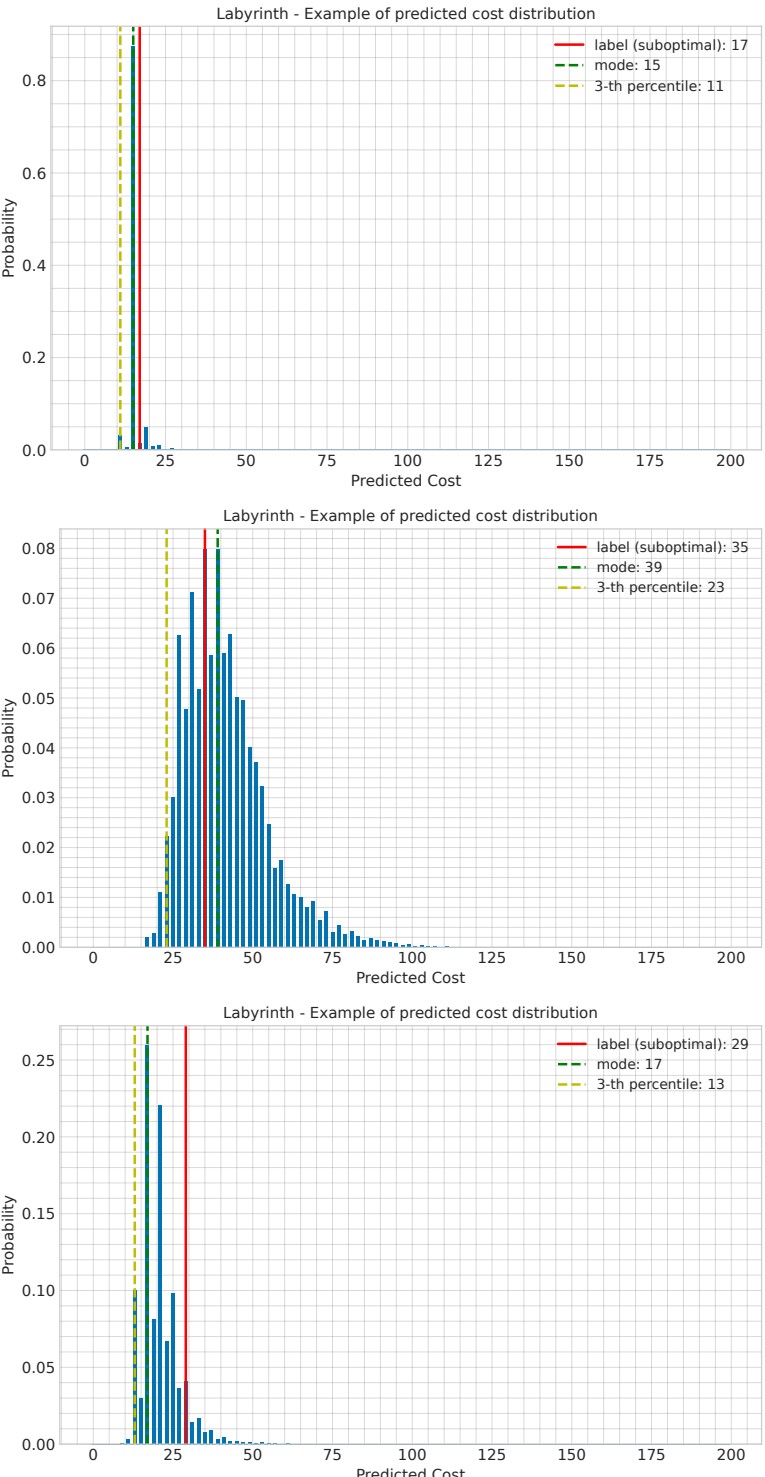

*Figure 10.* Labyrinth - Examples of cost distribution generated by the learned heuristic model.

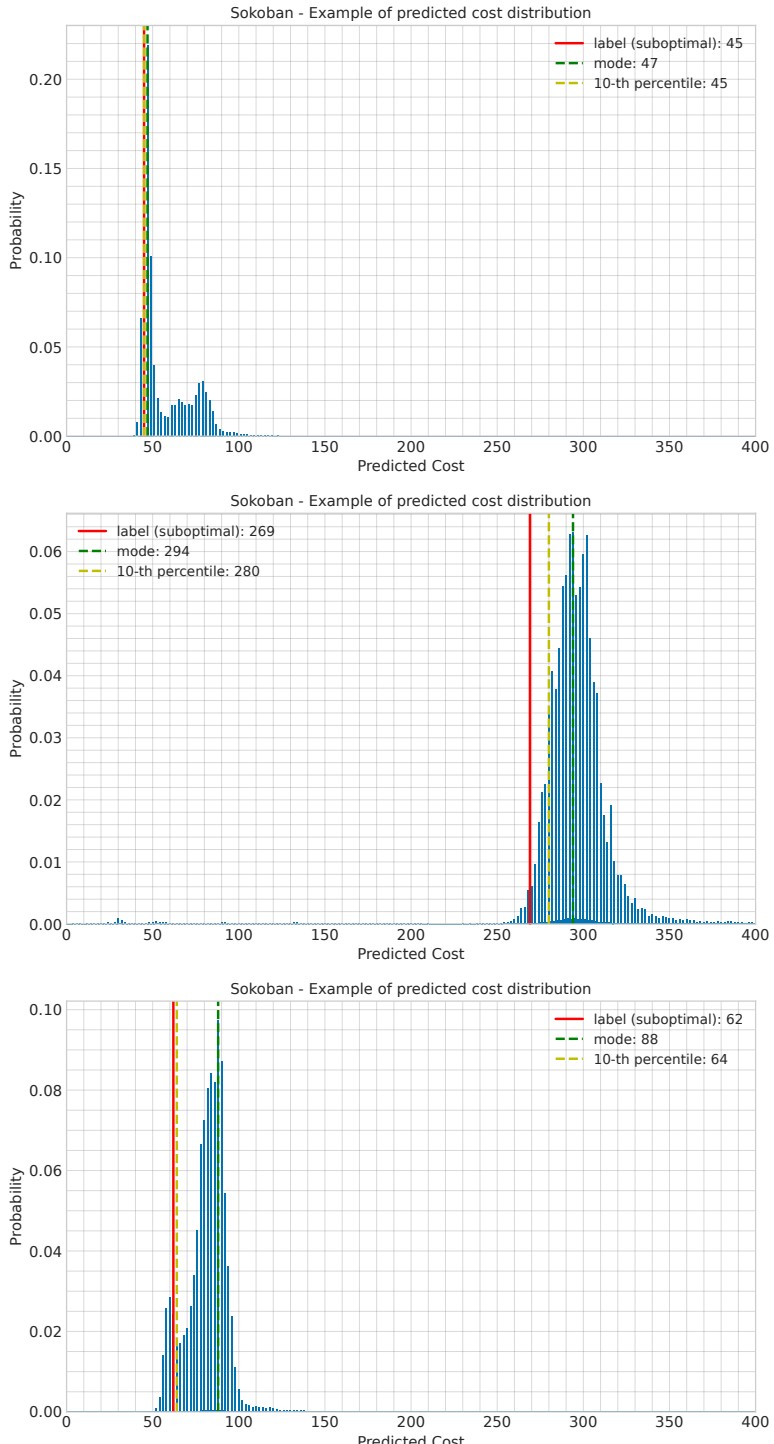

*Figure 11.* Sokoban - Examples of cost distribution generated by the learned heuristic model.

### F.4. Heuristic model vs. true optimal cost-to-go

We compare the predictions of our learned heuristic model against the true optimal cost-to-go for the subset of solved problems with known optimal solutions. Table 15 reports the mean absolute error (MAE) and mean error (ME; prediction − ground truth) statistics across all domains. We evaluate both the predicted heuristic value with the highest probability (Mode) and the value obtained through percentile estimation (Perc.).

|     |       | BW | BW ($n_{\text{loop}}$=3) | Log | Lab | Sok | Sok ($n_{\text{loop}}$=3) |
|-----|-------|------|------|------|------|------|------|
| MAE | Mode  | $8.68 \pm 0.24$ | $0.44 \pm 0.02$ | $3.41 \pm 0.09$ | $2.32 \pm 0.10$ | $5.28 \pm 0.17$ | $3.05 \pm 0.11$ |
|     | Perc. | $4.07 \pm 0.12$ | $0.66 \pm 0.02$ | $3.07 \pm 0.08$ | $1.07 \pm 0.04$ | $3.31 \pm 0.12$ | $2.51 \pm 0.08$ |
| ME  | Mode  | $8.68 \pm 0.24$ | $0.21 \pm 0.02$ | $1.40 \pm 0.11$ | $2.11 \pm 0.10$ | $4.79 \pm 0.17$ | $1.81 \pm 0.12$ |
|     | Perc. | $3.99 \pm 0.12$ | $-0.51 \pm 0.02$ | $-0.56 \pm 0.10$ | $0.56 \pm 0.05$ | $2.23 \pm 0.13$ | $-0.48 \pm 0.09$ |

*Table 15.* MAE (mean absolute error) and ME (mean error) statistics ($\pm$ std. error) with respect to the subset of known optimal data. Shown for both mode and percentile-based heuristic estimation across domains.

These results confirm the benefit of our percentile-based approach, which consistently shifts the predictions of the base heuristic model closer to the true optimal costs. Moreover, the self-improvement loop ($n_{\text{loop}} = 3$) further reduces the estimation error of both configurations, substantially improving overall heuristic accuracy.

### F.5. Model Generalization to Larger Number of Objects

Sec. 5.2 shows that OCLGEN generates valid plans where FD-LAMA failed, and outperforms Best-of-N sampling. Generalization beyond trained object counts is limited by the generative model's token vocabulary, which is a limitation shared by all baselines, not specific to our inference method.

Yet, we ran an additional experiment: training on Blocksworld with 3–25 blocks while excluding problems with 11, 12, 18, or 19 blocks, then testing on those (interpolate) and testing on instances with 26 to 40 blocks (extrapolate). The model with Best-of-N solved 100% of interpolation instances but 0% of extrapolation (unseen object names). The results on extrapolation are not surprising since the inputs during testing contain unobserved object name tokens (e.g. block31). In order to improve generalization to unseen object names, we retrained the base model while augmenting the data by randomly shuffling object names during the batch generation. With this new generative model, Best-of-N achieved 74.0% completion on the extrapolation set within a 10min time limit per problem. Test-time search with OCLGEN further improved the completion rate to 79.9%.

### F.6. Statistical Analysis

Table 1 in Section 5 presents the mean plan length and standard errors of all methods across different domains. Yet, in order to determine statistical significance, simply analyzing these statistics alone is insufficient since we test on the same 1000 instances per domain where problem difficulty dominates the variance. For example, the standard deviation of plan length is close to 110 on the Logistics test set. This shared structure means measurements are highly correlated across methods ($\rho > 0.81$, often $> 0.99$), so unpaired comparisons vastly inflate the standard error and mask consistent improvements. For example, in Logistics the unpaired $SE$ of the difference is $4.94$, while the paired $SE$ is just $0.11$ — a $45\times$ reduction.

We therefore perform Wilcoxon signed-rank tests on paired per-problem differences $\Delta_i = L_{\text{OCLGen},i} - L_{\text{baseline},i}$, comparing OCLGen against MCTS (full rollouts), the strongest baseline with comparable completion rates:

| Domain | $N$ | Mean $\Delta$ | Cohen's $d$ | Wilcoxon $p$ |
|--------|-----|------|------|------|
| Blocksworld | 1000 | $-10.68 \pm 0.31$ | $-1.08$ | $5.4 \cdot 10^{-136}$ |
| Logistics | 1000 | $-2.92 \pm 0.11$ | $-0.81$ | $1.4 \cdot 10^{-110}$ |
| Labyrinth | 1000 | $-2.82 \pm 0.15$ | $-0.60$ | $1.0 \cdot 10^{-66}$ |
| Sokoban | 998 | $-2.55 \pm 0.21$ | $-0.39$ | $3.2 \cdot 10^{-48}$ |

*Table 16.* Statistical comparison of OCLGEN vs. MCTS across domains.

Table 17 extends this comparison to a self-improvement setting, where both methods undergo 3 iterations of self-improvement

using their respective internal search strategies (OCLGen vs. MCTS).

| Domain | $N$ | Mean $\Delta$ | Cohen's $d$ | Wilcoxon $p$ |
|---|---|---|---|---|
| Blocksworld | 1000 | $-4.15 \pm 0.19$ | $-0.69$ | $4.1 \times 10^{-89}$ |
| Sokoban | 992 | $-2.08 \pm 0.13$ | $-0.51$ | $2.6 \times 10^{-56}$ |

*Table 17.* Statistical comparison of OCLGEN vs. MCTS (self-improvement, 3 iterations) across domains.

Across all domains and both experimental settings, the Wilcoxon signed-rank test returns p-values that are vanishingly small, ranging from $5.4 \times 10^{-136}$ in Blocksworld to $3.2 \times 10^{-48}$ in Sokoban (Table 16). All results comfortably exceed any conventional significance threshold (e.g., $\alpha = 0.05$ or even $\alpha = 0.001$). The self-improvement setting (Table 17) confirms this pattern, with p-values of $4.1 \times 10^{-89}$ and $2.6 \times 10^{-56}$ for Blocksworld and Sokoban respectively.

The Cohen's $d$ values provide a far more informative picture of OCLGen's advantage. Using the conventional benchmarks ($|d| = 0.2$ small, $|d| = 0.5$ medium, $|d| = 0.8$ large), the results reveal a consistent but domain-dependent advantage for OCLGen.

In Table 16, Blocksworld shows the strongest effect ($d = -1.08$), a large effect by any standard, accompanied by the largest absolute mean difference of $-10.68 \pm 0.31$ steps. Logistics follows with a large effect ($d = -0.81$, $\Delta = -2.92 \pm 0.11$), reflecting a similar advantage. Labyrinth yields a medium-to-large effect ($d = -0.60$, $\Delta = -2.82 \pm 0.15$), while Sokoban shows a smaller effect ($d = -0.39$, $\Delta = -2.55 \pm 0.21$), falling in the small-to-medium range.

In Table 17, both methods undergo 3 iterations of refinement using their own internal strategy before being compared head-to-head. OCLGen maintains a meaningful advantage over self-improved MCTS in both tested domains. Blocksworld yields $d = -0.69$ (medium-to-large) with $\Delta = -4.15 \pm 0.19$, and Sokoban yields $d = -0.51$ (medium) with $\Delta = -2.08 \pm 0.13$.

Taken together, the results demonstrate that OCLGen consistently and substantially outperforms MCTS in plan length across all tested domains. The statistical evidence is unambiguous, and the effect sizes range from practically meaningful ($d \approx -0.39$ in Sokoban) to large ($d \approx -1.08$ in Blocksworld). When both methods are given equal opportunity to self-improve over 3 iterations, OCLGen retains its advantage with medium-to-large effects in both domains.

# G. PDDL Domain files

## G.1. Blocksworld

*Listing 1.* Blocksworld domain file

```
(define (domain blocksworld-4ops)
 (:requirements :strips)
(:predicates (clear ?x)
         (on-table ?x)
         (arm-empty)
         (holding ?x)
         (on ?x ?y))

(:action pickup
 :parameters (?ob)
 :precondition (and (clear ?ob) (on-table ?ob) (arm-empty))
 :effect (and (holding ?ob) (not (clear ?ob)) (not (on-table ?ob))
         (not (arm-empty))))

(:action putdown
 :parameters (?ob)
 :precondition (holding ?ob)
 :effect (and (clear ?ob) (arm-empty) (on-table ?ob)
         (not (holding ?ob))))

(:action stack
 :parameters (?ob ?underob)
 :precondition (and (clear ?underob) (holding ?ob))
 :effect (and (arm-empty) (clear ?ob) (on ?ob ?underob)
         (not (clear ?underob)) (not (holding ?ob))))

(:action unstack
 :parameters (?ob ?underob)
 :precondition (and (on ?ob ?underob) (clear ?ob) (arm-empty))
 :effect (and (holding ?ob) (clear ?underob)
         (not (on ?ob ?underob)) (not (clear ?ob)) (not (arm-empty)))))
```

## G.2. Logistics

*Listing 2.* Logistics domain file

```
(define (domain logistics)
 (:requirements :strips :typing)
 (:types
     package location vehicle - object
     truck airplane - vehicle
     city airport - location)

 (:predicates
          (at ?vehicle-or-package - (either vehicle package) ?location - location)
          (in ?package - package ?vehicle - vehicle)
          (in-city ?loc-or-truck - (either location truck) ?citys - city))

 (:action load-truck
     :parameters
          (?obj - package
           ?truck - truck
           ?loc - location)
     :precondition
          (and  (at ?truck ?loc)
                (at ?obj ?loc))
     :effect
          (and  (not (at ?obj ?loc))
                (in ?obj ?truck)))

 (:action load-airplane
     :parameters
          (?obj - package
           ?airplane - airplane
           ?loc - airport)
     :precondition
          (and
                (at ?obj ?loc)
                (at ?airplane ?loc))
     :effect
          (and  (not (at ?obj ?loc))
                (in ?obj ?airplane)))

 (:action unload-truck
     :parameters
          (?obj - package
           ?truck - truck
           ?loc - location)
     :precondition
          (and (at ?truck ?loc)
                (in ?obj ?truck))
     :effect
          (and (not (in ?obj ?truck))
                (at ?obj ?loc)))

 (:action unload-airplane
     :parameters
          (?obj - package
           ?airplane - airplane
           ?loc - airport)
     :precondition
          (and (in ?obj ?airplane)
                (at ?airplane ?loc))
     :effect
          (and
                (not (in ?obj ?airplane))
                (at ?obj ?loc)))
```

```
(:action drive-truck
    :parameters
          (?truck - truck
           ?loc-from - location
           ?loc-to - location
           ?city - city)
    :precondition
          (and  (at ?truck ?loc-from)
                (in-city ?loc-from ?city)
                (in-city ?loc-to ?city))
    :effect
          (and  (not (at ?truck ?loc-from))
                (at ?truck ?loc-to)))

(:action fly-airplane
    :parameters
          (?airplane - airplane
           ?loc-from - airport
           ?loc-to - airport)
    :precondition
          (at ?airplane ?loc-from)
    :effect
          (and  (not (at ?airplane ?loc-from))
          (at ?airplane ?loc-to)))
)
```

## G.3. Labyrinth

*Listing 3.* Labyrinth domain file

```
(define (domain labyrinth)
(:requirements :adl :action-costs :strips :typing :numeric-fluents)

(:types
   card - object
   direction - object
   directionV - direction
   directionH - direction
   gridpos - object
)

(:constants
   s n - directionV
   w e - directionH
)

(:predicates
   (next ?p1 - gridpos ?p2 - gridpos)
   (max-pos ?p - gridpos)
   (min-pos ?p - gridpos)
   (blocked ?c - card ?d - direction)
   (robot-at ?c - card)
   (card-at ?c - card ?x - gridpos ?y - gridpos)
   (left)
   (cards-moving)
   (cards-moving-west)
   (cards-moving-east)
   (cards-moving-south)
   (cards-moving-north)
   (next-moving-card ?c - card)
   (new-headtail-card ?c - card)

)

(:functions
   (total-cost) - number
   (move-robot-cost) - number
   (move-card) - number
)

(:action move-west
   :parameters (?cfrom - card ?xfrom - gridpos ?yfrom - gridpos ?dfrom - directionH ?cto
      ↪ - card ?xto - gridpos ?yto - gridpos ?dto - directionH)
   :precondition
      (and
         (not (cards-moving))
         (= ?dfrom w)
         (robot-at ?cfrom)
         (card-at ?cfrom ?xfrom ?yfrom)
         (card-at ?cto ?xto ?yto)
         (next ?xfrom ?xto)
         (= ?yfrom ?yto)
         (not (= ?dfrom ?dto))
         (not (blocked ?cfrom ?dfrom))
         (not (blocked ?cto ?dto))
      )
   :effect
      (and
         (not (robot-at ?cfrom))
         (robot-at ?cto)
         (increase (total-cost) (move-robot-cost))
```

```
        )
)

(:action move-east
    :parameters (?cfrom - card ?xfrom - gridpos ?yfrom - gridpos ?dfrom - directionH ?cto
        ↪ - card ?xto - gridpos ?yto - gridpos ?dto - directionH)
    :precondition
        (and
            (not (cards-moving))
            (= ?dfrom e)
            (robot-at ?cfrom)
            (card-at ?cfrom ?xfrom ?yfrom)
            (card-at ?cto ?xto ?yto)
            (next ?xto ?xfrom)
            (= ?yfrom ?yto)
            (not (= ?dfrom ?dto))
            (not (blocked ?cfrom ?dfrom))
            (not (blocked ?cto ?dto))
        )
    :effect
        (and
            (not (robot-at ?cfrom))
            (robot-at ?cto)
            (increase (total-cost) (move-robot-cost))
        )
)

(:action move-north
    :parameters (?cfrom - card ?xfrom - gridpos ?yfrom - gridpos ?dfrom - directionV ?cto
        ↪ - card ?xto - gridpos ?yto - gridpos ?dto - directionV)
    :precondition
        (and
            (not (cards-moving))
            (= ?dfrom n)
            (robot-at ?cfrom)
            (card-at ?cfrom ?xfrom ?yfrom)
            (card-at ?cto ?xto ?yto)
            (next ?yfrom ?yto)
            (= ?xfrom ?xto)
            (not (= ?dfrom ?dto))
            (not (blocked ?cfrom ?dfrom))
            (not (blocked ?cto ?dto))
        )
    :effect
        (and
            (not (robot-at ?cfrom))
            (robot-at ?cto)
            (increase (total-cost) (move-robot-cost))
        )
)

(:action move-south
    :parameters (?cfrom - card ?xfrom - gridpos ?yfrom - gridpos ?dfrom - directionV ?cto
        ↪ - card ?xto - gridpos ?yto - gridpos ?dto - directionV)
    :precondition
        (and
            (not (cards-moving))
            (= ?dfrom s)
            (robot-at ?cfrom)
            (card-at ?cfrom ?xfrom ?yfrom)
            (card-at ?cto ?xto ?yto)
            (next ?yto ?yfrom)
            (= ?xfrom ?xto)
            (not (= ?dfrom ?dto))
            (not (blocked ?cfrom ?dfrom))
```

```
            (not (blocked ?cto ?dto))
        )
    :effect
        (and
            (not (robot-at ?cfrom))
            (robot-at ?cto)
            (increase (total-cost) (move-robot-cost))
        )
)

(:action start-move-card-west
:parameters(?cm - card ?x - gridpos ?y - gridpos ?cnext - card ?nextx - gridpos)
:precondition
    (and
        (not (cards-moving))
        (not (cards-moving-west))
        (not (robot-at ?cm))
        (card-at ?cm ?x ?y )
        (min-pos ?x)
        (card-at ?cnext ?nextx ?y)
        (next ?nextx ?x)
    )
:effect
    (and
        (cards-moving)
        (cards-moving-west)
        (not (card-at ?cm ?x ?y ))
        (new-headtail-card ?cm)
        (next-moving-card ?cnext)
        (increase (total-cost) (move-card))
    )
)

(:action move-card-west
:parameters(?cm - card ?x - gridpos ?y - gridpos ?cnext - card ?nextx - gridpos ?prevx -
    ↪ gridpos)
:precondition
    (and
        (cards-moving)
        (cards-moving-west)
        (not (robot-at ?cm))
        (next-moving-card ?cm)
        (card-at ?cm ?x ?y )
        (card-at ?cnext ?nextx ?y)
        (next ?x ?prevx)
        (next ?nextx ?x)
    )
:effect
    (and
        (cards-moving)
        (cards-moving-west)
        (not (card-at ?cm ?x ?y))
        (card-at ?cm ?prevx ?y)
        (not (next-moving-card ?cm))
        (next-moving-card ?cnext)
        (increase (total-cost) (move-card))
    )
)

(:action stop-move-card-west
:parameters(?cm - card ?x - gridpos ?y - gridpos ?prevx - gridpos ?newtc - card)
:precondition
    (and
        (cards-moving)
        (cards-moving-west)
```

```
            (not (robot-at ?cm))
            (next-moving-card ?cm)
            (card-at ?cm ?x ?y )
            (next ?x ?prevx)
            (max-pos ?x)
            (new-headtail-card ?newtc)
      )
:effect
      (and
            (not (cards-moving))
            (not (cards-moving-west))
            (not (card-at ?cm ?x ?y))
            (card-at ?cm ?prevx ?y)
            (card-at ?newtc ?x ?y)
            (not (new-headtail-card ?newtc))
            (not (next-moving-card ?cm))
            (increase (total-cost) (move-card))
      )
)

(:action start-move-card-east
:parameters(?cm - card ?x - gridpos ?y - gridpos ?cnext - card ?nextx - gridpos)
:precondition
      (and
            (not (cards-moving))
            (not (cards-moving-east))
            (not (robot-at ?cm))
            (card-at ?cm ?x ?y )
            (max-pos ?x)
            (card-at ?cnext ?nextx ?y)
            (next ?x ?nextx)
      )
:effect
      (and
            (cards-moving)
            (cards-moving-east)
            (not (card-at ?cm ?x ?y ))
            (new-headtail-card ?cm)
            (next-moving-card ?cnext)
            (increase (total-cost) (move-card))
      )
)

(:action move-card-east
:parameters(?cm - card ?x - gridpos ?y - gridpos ?cnext - card ?nextx - gridpos ?prevx -
      ↪ gridpos)
:precondition
      (and
            (cards-moving)
            (cards-moving-east)
            (not (robot-at ?cm))
            (next-moving-card ?cm)
            (card-at ?cm ?x ?y )
            (card-at ?cnext ?nextx ?y)
            (next ?prevx ?x)
            (next ?x ?nextx)
      )
:effect
      (and
            (cards-moving)
            (cards-moving-east)
            (not (card-at ?cm ?x ?y))
            (card-at ?cm ?prevx ?y)
            (not (next-moving-card ?cm))
            (next-moving-card ?cnext)
```

```
            (increase (total-cost) (move-card))
    )
)

(:action stop-move-card-east
:parameters(?cm - card ?x - gridpos ?y - gridpos ?prevx - gridpos ?newtc - card)
:precondition
    (and
        (cards-moving)
        (cards-moving-east)
        (not (robot-at ?cm))
        (next-moving-card ?cm)
        (card-at ?cm ?x ?y )
        (next ?prevx ?x)
        (min-pos ?x)
        (new-headtail-card ?newtc)
    )
:effect
    (and
        (not (cards-moving))
        (not (cards-moving-east))
        (not (card-at ?cm ?x ?y))
        (card-at ?cm ?prevx ?y)
        (card-at ?newtc ?x ?y)
        (not (new-headtail-card ?newtc))
        (not (next-moving-card ?cm))
        (increase (total-cost) (move-card))
    )
)

(:action start-move-card-north
:parameters(?cm - card ?x - gridpos ?y - gridpos ?cnext - card ?nexty - gridpos)
:precondition
    (and
        (not (cards-moving))
        (not (cards-moving-north))
        (not (robot-at ?cm))
        (card-at ?cm ?x ?y )
        (min-pos ?y)
        (card-at ?cnext ?x ?nexty)
        (next ?nexty ?y)
    )
:effect
    (and
        (cards-moving)
        (cards-moving-north)
        (not (card-at ?cm ?x ?y ))
        (new-headtail-card ?cm)
        (next-moving-card ?cnext)
        (increase (total-cost) (move-card))
    )
)

(:action move-card-north
:parameters(?cm - card ?x - gridpos ?y - gridpos ?cnext - card ?nexty - gridpos ?prevy -
    ↪ gridpos)
:precondition
    (and
        (cards-moving)
        (cards-moving-north)
        (not (robot-at ?cm))
        (next-moving-card ?cm)
        (card-at ?cm ?x ?y )
        (card-at ?cnext ?x ?nexty)
        (next ?y ?prevy)
```

```
        (next ?nexty ?y)
    )
:effect
    (and
        (cards-moving)
        (cards-moving-north)
        (not (card-at ?cm ?x ?y))
        (card-at ?cm ?x ?prevy)
        (not (next-moving-card ?cm))
        (next-moving-card ?cnext)
        (increase (total-cost) (move-card))
    )
)

(:action stop-move-card-north
:parameters(?cm - card ?x - gridpos ?y - gridpos ?prevy - gridpos ?newtc - card)
:precondition
    (and
        (cards-moving)
        (cards-moving-north)
        (not (robot-at ?cm))
        (next-moving-card ?cm)
        (card-at ?cm ?x ?y )
        (next ?y ?prevy)
        (max-pos ?y)
        (new-headtail-card ?newtc)
    )
:effect
    (and
        (not (cards-moving))
        (not (cards-moving-north))
        (not (card-at ?cm ?x ?y))
        (card-at ?cm ?x ?prevy)
        (card-at ?newtc ?x ?y)
        (not (new-headtail-card ?newtc))
        (not (next-moving-card ?cm))
        (increase (total-cost) (move-card))
    )
)

(:action start-move-card-south
:parameters(?cm - card ?x - gridpos ?y - gridpos ?cnext - card ?nexty - gridpos)
:precondition
    (and
        (not (cards-moving))
        (not (cards-moving-south))
        (not (robot-at ?cm))
        (card-at ?cm ?x ?y )
        (max-pos ?y)
        (card-at ?cnext ?x ?nexty)
        (next ?y ?nexty)
    )
:effect
    (and
        (cards-moving)
        (cards-moving-south)
        (not (card-at ?cm ?x ?y ))
        (new-headtail-card ?cm)
        (next-moving-card ?cnext)
        (increase (total-cost) (move-card))
    )
)

(:action move-card-south
:parameters(?cm - card ?x - gridpos ?y - gridpos ?cnext - card ?nexty - gridpos ?prevy -
```

```
     ↪ gridpos)
:precondition
    (and
        (cards-moving)
        (cards-moving-south)
        (not (robot-at ?cm))
        (next-moving-card ?cm)
        (card-at ?cm ?x ?y )
        (card-at ?cnext ?x ?nexty)
        (next ?prevy ?y)
        (next ?y ?nexty)
    )
:effect
    (and
        (cards-moving)
        (cards-moving-south)
        (not (card-at ?cm ?x ?y))
        (card-at ?cm ?x ?prevy)
        (not (next-moving-card ?cm))
        (next-moving-card ?cnext)
        (increase (total-cost) (move-card))
    )
)

(:action stop-move-card-south
:parameters(?cm - card ?x - gridpos ?y - gridpos ?prevy - gridpos ?newtc - card)
:precondition
    (and
        (cards-moving)
        (cards-moving-south)
        (not (robot-at ?cm))
        (next-moving-card ?cm)
        (card-at ?cm ?x ?y )
        (next ?prevy ?y)
        (min-pos ?y)
        (new-headtail-card ?newtc)
    )
:effect
    (and
        (not (cards-moving))
        (not (cards-moving-south))
        (not (card-at ?cm ?x ?y))
        (card-at ?cm ?x ?prevy)
        (card-at ?newtc ?x ?y)
        (not (new-headtail-card ?newtc))
        (not (next-moving-card ?cm))
        (increase (total-cost) (move-card))
    )
)

(:action leave
:parameters(?c - card ?prow - gridpos ?pcolumn - gridpos)
:precondition
    (and
        (not (cards-moving))
        (robot-at ?c)
        (card-at ?c ?prow ?pcolumn)
        (max-pos ?prow)
        (max-pos ?pcolumn)
        (not (blocked ?c s ))
    )
:effect
    (and
        (increase (total-cost) (move-card))
        (left)
```

```
        )
)
)
```

## G.4. Sokoban

*Listing 4.* Sokoban domain file

```
(define (domain typed-sokoban)
(:requirements :typing)
(:types LOC DIR)
(:predicates
        (at-robot ?l - LOC)
        (has-box ?l - LOC)
        (adjacent ?l1 - LOC ?l2 - LOC ?d - DIR)
        (clear ?l - LOC)
)

(:action move
:parameters (?from - LOC ?to - LOC ?dir - DIR)
:precondition (and (clear ?to) (at-robot ?from) (adjacent ?from ?to ?dir))
:effect (and (at-robot ?to) (not (at-robot ?from)))
)

(:action push
:parameters (?rloc - LOC ?bloc - LOC ?floc - LOC ?dir - DIR)
:precondition (and (at-robot ?rloc) (has-box ?bloc) (clear ?floc)
            (adjacent ?rloc ?bloc ?dir) (adjacent ?bloc ?floc ?dir))

:effect (and (at-robot ?bloc) (has-box ?floc) (clear ?bloc)
        (not (at-robot ?rloc)) (not (has-box ?bloc)) (not (clear ?floc)))
)
```

