# OpenReview forum: "Efficient Test-time Inference for Generative Planning Models with OCL Search"
_ICML.cc/2026/Conference — ICML 2026 regular_

### Official Review · Reviewer_4pj7 · 2026-03-05

**Soundness:** 3
**Presentation:** 3
**Significance:** 2
**Originality:** 2
**Overall Recommendation:** 4
**Confidence:** 4

**Summary:**

The paper proposes an anytime tree search algorithm guided by neural networks for solving classical planning problems (deterministic fully observed planning; they experiment on PDDL problems). The high level idea is to train models that serve as heuristics and rollout policies based on training data from suboptimal planners. The actual tree search algorithm itself resembles MCTS in that it does rollouts, but resembles more classical tree search algorithms in that it maintains open/closed lists of explored nodes. Across 4 planning domains the method outperform baselines. There are several nonstandard (to me) ideas such as systematically sweeping through nodes of different depth, and "snapping" the neural heuristic estimates to lower values (which compensates for having trained the heuristic on suboptimal plans, hence the heuristic distance would generally otherwise overestimate). They validate these ideas via ablations. They also show that the system can be used to generate its own training data, in the style of expert iteration.

**Compliance With Llm Reviewing Policy:**

Affirmed.

**Final Justification:**

I've raised my score in response to the rebuttal:
1. The method seems more robust to hyperparameter selection than the initial submission suggested. I'm still little confused here because the initial submission made it seem like one of the benchmarks required (very) different hyerparameters but the window for discussion has closed and I don't want to penalize the authors here.
2. Effect size is not huge in plan length, but great in percentage of plans that are optimal

I still think the method is adhoc and likely not close to the final form for this genre of search. I see the paper as a possible stepping stone to a less adhoc version of the algorithm. I hope if the paper is accepted the authors clarify *in the abstract* that they are only considering deterministic fully observed discrete decisionmaking problems.

**Key Questions For Authors:**

Are your empirical results statistically significant?

How robust is your method to misspecified hyperparameters?

**Limitations:**

yes, although I wish that they were more upfront earlier as mentioned in weaknesses

**Strengths And Weaknesses:**

Strengths:
- The paper is clear and well explained
- Although I have some criticisms of the method below, there were some creative decisions made in the algorithm, and if those decisions could prove valuable in more general decisionmaking settings (partially observed/stochastic) then I think there is merit to the paper
- The method solves some really hard planning problems! For example, Sokoban problems with >100 actions needed to solve them.

Weaknesses:

- I feel that the method is a bit strange and ad hoc, and therefore unlikely to be widely adopted. For example, sweeping through levels of the tree according to simple adhoc orderings, or tweaking the heuristic estimates downward
- These ad hoc decisions introduce extra hyper parameters. And even though only four planning domains are considered, the authors needed to make one of those hyper parameters very different in one of the domains.
- I was impressed by the Sokoban performance, but the baselines also seem to do ok, blunting the impact of the empirical result
- Limitations are only discussed much later, and should be foregrounded in the abstract. This is for deterministic fully observed planning problems. Many readers will go into this paper hoping to learn how to do planning for eg LLM agents and be disappointed.
- The section on self improvement has **no citations** -- surely this an oversight? Expert Iteration, AlphaGo, even TDGammon did this.
- I *think* the quantitative results are statistically significant, but I can't be sure because they give standard deviation instead of SEM or some other measure of error bars.

---

> ### Author Rebuttal · Authors · 2026-03-30
>
> We thank the reviewer for taking the time to read our paper and for the feedback.
>
> # 1. Merit of deterministic setting.
> Classical AI planning (deterministic/discrete) is an important branch of AI with its own international competitions [https://www.icaps-conference.org/competitions/]. Our method uses PDDL as the formal language: the token vocabulary is generated directly from PDDL, making it applicable to combinatorially-hard planning and scheduling problems in industry (logistics, robot fleet management, ..). We believe there is clear merit in algorithms that push search capabilities further in this established setting.
>
> # 2. "Ad hoc" design choices.
> We believe algorithmic simplicity is an advantage, making it easier for others to adapt. In additions to the motivations in Sec 4.1-4.4, we would gladly use additional space to expand the theoretical motivation behind each component (see W1 for reviewer JorU).
>
> # 3. Baselines seem to do ok.
> Our focus lies on solution quality (plan length/optimality), not just completion. Finding optimal plans is significantly harder than finding any solution (e.g., optimal Blocksworld is NP-hard while feasible solutions are polynomial). Since all baselines share the same base model, high completion rates are expected; the key metrics are plan length and optimality ratio. OCLGen significantly outperforms MCTS even in Sokoban ($p < 10^{-48}$, see below), and the improvement is consistent across *all* domains. Baselines like GBFS and Anyt. A* fail badly in Logistics and Sokoban, highlighting that strong performance across all domains is non-trivial.
>
> # 4. Hyperparam. differences.
> We selected $\tau_\text{conf}$ and $k$ on a small validation set from $\tau_\text{conf} \in$ {0.2, 0.5, 0.95} and $k \in$ {3, 10}. We added an ablation of these parameters on the test set. Please see results in Table under 8. below. While Logistics favored a different $\tau_\text{conf}$, performance remains nearly identical across settings and still outperforms baselines. The rollout length (50 vs. 2000 tokens) was initially chosen to reflect differences in plan/action-description length across domains, rather than sensitive tuning.
>
> # 5. Mention determinism early.
> While we already refer to "AI/automated planning"(typically assumes determinism) throughout the abstract/introduction, we agree and are happy to explicitly state the assumption more prominently in the introduction.
>
> # 6. Self-improvement citations.
> Thank you for suggesting this. We did not include a dedicated self-impr. background section due to our main focus on single-inference performance and the given page limit. We will add a new paragraph to the background with relevant references, e.g., Silver et al. (AlphaZero), Anthony et al. (ExIt), Schrittwieser et al. (MuZero), among others.
>
> # 7. Statistical significance.
> We report Table 1 with std. errors (SE):
>
> | | BW | Log | Lab | Sok |
> |-|-|-|-|-|
> | **OCLGen** (uniform) | 43.88 ± 0.68 | 155.83 ± 3.47 | 12.99 ± 0.13 | 128.67 ± 3.15 |
> | **OCLGen** (scan) | 44.10 ± 0.70 | 157.63 ± 3.51 | 13.00 ± 0.13 | 128.10 ± 3.15 |
> | MCTS (full) | 54.56 ± 0.93 | 158.75 ± 3.52 | 15.81 ± 0.23 | 131.30 ± 3.23 |
> | MCTS (partial) | 53.81 ± 0.90 | 40.31 ± 1.30 | 14.77 ± 0.19 | 38.34 ± 0.93 |
> | OCL-Anyt. A* | 37.07 ± 0.71 | 2.81 ± 0.70 | 13.04 ± 0.13 | 37.30 ± 1.11 |
> | OCL-GBFS | 65.48 ± 1.50 | 2.46 ± 0.85 | 18.03 ± 0.43 | 48.38 ± 1.72 |
> | Best-of-N | 61.56 ± 1.03 | 157.00 ± 3.47 | 17.42 ± 0.25 | 132.25 ± 3.22 |
>
> All methods are evaluated on the same 1k problems and measurements are highly correlated ($\rho > 0.81$, often $> 0.99$). As unpaired comparisons ignore this correlation and vastly inflate the standard error of the difference, comparing overlapping SE (Table 1) is not appropriate for assessing statistical significance. We instead perform **Wilcoxon signed-rank tests** on paired differences $\Delta_i = L_{\text{OCLGen},i} - L_{\text{baseline},i}$, comparing OCLGen (unif.) against MCTS (full), the strongest baseline with comparable completion.
>
> | Domain | Mean $\Delta$ | Wilcoxon $p$ |
> |-|-|-|
> | Blocksworld | −10.68 ± 0.31 | 5.4 × 10⁻¹³⁶ |
> | Logistics | −2.92 ± 0.11 | 1.4 × 10⁻¹¹⁰ |
> | Labyrinth | −2.81 ± 0.15 | 1.0 × 10⁻⁶⁶ |
> | Sokoban | −2.55 ± 0.21 | 3.2 × 10⁻⁴⁸ |
>
> This proves that OCLGen produces **significantly shorter plans** across domains ($p < 10^{-48}$).
>
> # 8. Hyperparam. robustness (also see item 4 above).
> $\tau_\text{conf}$ sweep ($k$ fixed):
>
> | | BW (0.2) | BW (0.5) | BW (0.95) | Log (0.2) | Log (0.5) | Log (0.95) |
> |-|-|-|-|-|-|-|
> | Plan Len. | 44.26 ± 21.88 | 44.19 ± 21.93 | 43.88 ± 21.35 | 155.83 ± 109.68 | 155.76 ± 109.72 | 155.95 ± 109.84 |
>
> $k$ sweep ($\tau_\text{conf}$ fixed):
>
> | | BW (3) | BW (10) | Log (3) | Log (10) |
> |-|-|-|-|-|
> | Plan Len. | 43.88 ± 21.35 | 44.05 ± 21.53 | 155.84 ± 109.65 | 155.83 ± 109.68 |
>
> Performance is nearly identical, confirming robustness to hyperparameter choice.
>
> We welcome any remaining questions and respectfully ask the reviewer to reconsider their score.

---

> > ### Author Rebuttal · Reviewer_4pj7 · 2026-04-03
> >
> > Thanks!
> >
> > Can you discuss both effect size and statistical significance? Seems it's consistently 1-2% better (shorter) on sokoban vs MCTS? I think the issue is not just significance but effect size.
> >
> > Can you explain the hyper param sweep -- and for each hyper param setting, report overall normalized performance across all bencharks?
> >
> > I still see the method as ad-hoc.

---

> > > ### Author Response · Authors · 2026-04-05
> > >
> > > Thank you for the additional questions. We ran further experiments which we believe strengthen the significance of our contribution and hope they address the reviewer's concerns.
> > >
> > > # 1. Paired test: OCLGen vs. MCTS
> > >
> > > Wilcoxon signed-rank test is used as the primary significance test and Cohen's $d$ as the paired effect size measure. A negative mean difference Δ indicates that OCLGen produces shorter plans. The table below shows our result:
> > >
> > > ||Δ|Cohen's d|p|
> > > |-|-|-|-|
> > > |BW|-10.68 ± 0.31|-1.08|$5.4 \times 10^{-136}$|
> > > |Log|-2.92 ± 0.11|-0.81|$1.4 \times 10^{-110}$|
> > > |Lab|-2.81 ± 0.15|-0.60|$1.0 \times 10^{-66}$|
> > > |Sok|-2.55 ± 0.21|-0.39|$3.2 \times 10^{-48}$|
> > >
> > > We extend this comparison to the self-improvement setting (Table 6; OCLGen vs. MCTS):
> > >
> > > ||Δ|Cohen's d|p|
> > > |-|-|-|-|
> > > |BW|-4.15 ± 0.19|-0.69|$4.1 \times 10^{-89}$|
> > > |Sok|-2.08 ± 0.13|-0.51|$2.6 \times 10^{-56}$|
> > >
> > > **Significance:** All p-values are vanishingly small (range: $3.2\times10^{-48}$ to $5.4\times10^{-136}$; self-improvement: $2.6\times10^{-56}$ to $4.1\times10^{-89}$), comfortably exceeding any conventional threshold (e.g., $\alpha = 0.05$).
> > >
> > > **Effect Sizes:** Using the conventional benchmarks ($|d|=0.2$ small, $|d|=0.5$ medium, $|d|=0.8$ large), the results reveal a consistent but domain-dependent advantage for OCLGen.
> > >
> > > Blocksworld shows the strongest effect ($d=-1.08$), a large effect by any standard. Logistics follows with a large effect ($d=-0.81$), reflecting a similar advantage. Labyrinth yields a medium-to-large effect ($d = -0.60$), while Sokoban shows a smaller effect ($d=-0.39$), falling in the small-to-medium range.
> > >
> > > In the self-improvement setting, OCLGen maintains a meaningful advantage: Blocksworld yields $d = -0.69$ (medium-to-large), and Sokoban yields $d=-0.51$ (medium).
> > >
> > > **Summary:** The results demonstrate that OCLGen consistently and substantially outperforms MCTS in plan length across all tested domains, with statistical evidence and effect sizes ranging from practically meaningful ($d\approx-0.39$) to large ($d\approx-1.08$).
> > >
> > > **Optimality** Plan optimality (%) is the second key metric (Table 2). OCLGen consistently yields a significantly higher % of optimal plans compared to all baselines, a trend that holds in the self-improvement experiments as well (Table 6). We show % of optimal plans in the table below:
> > >
> > > ||OCLGen|MCTS|
> > > |-|-|-|
> > > |BW|**83.81**|28.57|
> > > |Log|**61.54**|37.87|
> > > |Lab|**98.8**|60.5|
> > > |Sok|**77.10**|59.51|
> > >
> > > The numbers clearly support the dominance of our method in terms of %-optimal:
> > >
> > > # 2. Hyperparam sweep
> > >
> > > To address robustness to misspecified hyperparameters, we swept the two most critical ones, $\tau_\text{conf}$ and $k$. Given the substantial compute cost -- each configuration requires running 1000 test problems for 10 minutes each on a GPU -- we evaluated $\tau_\text{conf} \in$ {0.2, 0.5, 0.95} and $k \in$ {3, 10}, varying one at a time while keeping the other at its default. Results are shown below, with plan length and completion rate measured under an identical 10-min budget on the same hardware ("normalized").
> > >
> > > $\tau_\text{conf}$ (value in parentheses):
> > >
> > > ||Compl.|Plan Len.|
> > > |-|-|-|
> > > |BW (0.2)|100|44.26 ± 21.88|
> > > |BW (0.5)|100|44.19 ± 21.93|
> > > |BW (0.95)|100|43.88 ± 21.35|
> > > |Log (0.2)|100|155.83 ± 109.68|
> > > |Log (0.5)|100|155.76 ± 109.72|
> > > |Log (0.95)|100|155.95 ± 109.84|
> > > |Lab (0.2)|100|13.0 ± 4.01|
> > > |Lab (0.5)|100|12.99 ± 3.99|
> > > |Lab (0.95)|100|12.99 ± 3.98|
> > > |Sok (0.2)|99.8|128.66 ± 99.7|
> > > |Sok (0.5)|99.8|128.61 ± 99.83|
> > > |Sok (0.95)|99.9|128.67 ± 99.60|
> > >
> > > k (value in parentheses):
> > >
> > > ||Compl.|Plan Len.|
> > > |-|-|-|
> > > |BW (3)|100|43.88 ± 21.35|
> > > |BW (10)|100|44.05 ± 21.53|
> > > |Log (3)|100|155.84 ± 109.65|
> > > |Log (10)|100|155.83 ± 109.68|
> > > |Lab (3)|100|12.99 ± 3.98|
> > > |Lab (10)|100|13.01 ± 4.01|
> > > |Sok (3)|100|128.91 ± 99.92|
> > > |Sok (10)|99.9|128.67 ± 99.60|
> > >
> > > Across all domains and both sweeps, performance remains consistently strong, with only marginal variation. This indicates that our method is robust to reasonable deviations from the default hyperparameter choices.
> > >
> > > # 3. "Ad-hoc"
> > >
> > > OCLGen extends the classical OCL search by integrating fast generation with autoregressive models and learned heuristics. Key innovations include expansion via rollouts, adaptive branching, depth-separated open lists, and optimistic heuristic estimation from a distributional model. From a classical search perspective, rollouts reduce the effective branching factor, while depth-specific open lists counteract h-cost overestimation bias. Learning distributional heuristic models further enables percentile-based estimation to extract better h-cost estimates from suboptimal training data.
> > >
> > > We believe OCLGen constitutes a genuine contribution: unlike much prior work leveraging autoregressive models in search, which often relies on MCTS, our method is built on the OCL framework and demonstrates empirical dominance across challenging combinatorial planning domains. We further demonstrate its utility as a plan improvement operator in a recursive self-improvement framework.

---

### Official Review · Reviewer_7Gbc · 2026-03-11

**Soundness:** 3
**Presentation:** 2
**Significance:** 3
**Originality:** 3
**Overall Recommendation:** 4
**Confidence:** 4

**Summary:**

The paper proposes three techniques for improving the quality/length of classical plans, which make use of a learned stochastic policy for generating actions given states, and a learned heuristic appraisal function for evaluating states. The three techniques are depth-partitioned node selection, confidence-based node expansion, and a heuristic point estimator. The experimental results show that each of these three techniques make a significance difference in the overall performance, and that their combination achieves better results in plan quality than other baselines that also use the stochastic policy and/or the heuristic appraisal function.

**Compliance With Llm Reviewing Policy:**

Affirmed.

**Final Justification:**

The problem addressed in the paper -- improving plan quality with a pretty "good", learned greedy policy, is interesting and challlenging, and the techniques proposed manage to produce good results, better than what can be obtained by other methods, including MCTS. The contrasts, differences, and commonalities with MCTS and more generaly real-time vs. best-first search are also interesting and intriguing in this context, and hopefully can be explored more systematically in future work.

**Key Questions For Authors:**

1. The coverage results in table 1 for all the approaches that use the learned stochastic policy, including MCT and Best-of-N are very high; all these approaches appear to solve 100% of the instances. I've looked in the literature though, and I don't find papers that report policies that solve nearly 100% of the instances in Logistics or Sokoban. How is this possible? It's not claimed as a contribution of the paper, but I haven't seen this coverage elsewhere. Can you explain?

2. It'd be interesting to see the number of times in which the stochastic policy has been called on average in each domain, assuming that the policy is is called once when it results in full state trajectories being added to the open list. E.g., if I undersand correctly, best-of-N for N=1, call the stochastic policy once, but this single call may result in a complete plan if the resulting state trajectory reaches the goal.

**Limitations:**

Yes

**Strengths And Weaknesses:**

A strength of the paper is the proposed techniques appear to make a significant difference in the experimental results, showing a clear separation with the baselines, including MCTS and Best-of-N.

The main weakness of the paper in my view is that the three techniques, depth-partitioned selection, confidence-based node expansion, and distributional model and not explained with sufficient detail and precision. So, even if the the results are good, a reader would have difficulties in reproducing the code and the results. There are also a number of questions that pop up in my mind (see below).

The description of OCL search in section 4.1 (before the Modifications) looks too much like A*.

---

> ### Author Rebuttal · Authors · 2026-03-30
>
> Thank you for reading our manuscript and providing your detailed feedback. Please consider our response below. We welcome any further questions.
>
> # W1: Explanation of building blocks
>
> Thank you for pointing this out. While the character limit in our response doesn't allow us to post an updated description, we'd like to provide a more detailed description of the "depth-selection", "adaptive expansion" and "distributional heuristic" building blocks using the additional page allowed for the final version of the paper. Additionally we are happy to provide more detailed pseudocode in the appendix for each of the phases of the algorithm in Figure 1 in the paper, which should allow for better reproducibility of our method. Please also consider point W1 in our answer to Reviewer JorU where we provide more detail about the motivation of our method.
>
> # W2: "Similarity of OCL to A*"
>
> The reviewer is correct that OCL in its basic form resembles A*. This is intentional since OCL is a general framework for Open Closed List algorithms, and a large number of existing classical heuristic search algorithms can be seen as an instance of it, with A* being one particular case. We start from this well-understood search framework and introduce modifications (Sec. 4.1) that adapt it to work with auto-regressive generative models rather than heuristics only. Compared to A*, OCL does not necessarily assume an admissible heuristic and does not terminate at the first goal state found (i.e. anytime algorithm). One of our baselines, OCL-Anyt.-A* in Table 1, is an anytime variant of A*.
>
> # Q1: 100% solved instances in Logistics or Sokoban. How is this possible?
>
> Great question! It is indeed true that we solve almost 100% of unseen test problems in these domains. Our generative model is based on the architecture in PlanGPT [1]. If you look at the numbers in the PlanGPT paper, you see that they already achieve relatively high completion rates (e.g. blocksworld Table 1 in [1]), so our near-100% results build on an already strong baseline. At the same time we actually increased the problem complexity by using higher numbers of objects (e.g. up to 25 blocks instead of 20, or up to 50 cities in logistics instead of 10). There are multiple reasons for why we achieve better results on even harder problems:
>
> 1. As we report in Sec. 4.5, "Base Model Improvements via Action Compilation", we augment our data by generating and tokenizing intermediate state+plan pairs on-the-fly during batch computation during training.
> 2. We scaled up the base model training by using 100k instead of 63k training instances.
> 3. Our models are trained for a longer number of epochs.
>
> The largest improvements were achieved by scaling up training (items 2 and 3) which eventually led to the boost in performance of the base model compared to PlanGPT.
>
> While completion rates are high, the **main focus of the paper is to improve plan quality** (plan length/optimality), which is achieved not by scaling up training but by introducing our novel test-time search algorithm.
>
> [1] Learning General Policies for Planning through GPT Models, Rossetti et al. 2024
>
> # Q2: Number of generative model calls
>
> "best-of-N for N=1, call the stochastic policy once, but this single call may result in a complete plan..." This is correct, N=1 would mean querying the model once to generate a plan sequence. For N>1 we would generate multiple plans and report the shortest valid one out of those N. Note however that in order to achieve a fair comparison across methods we do not set N directly but instead define a maximum compute budget for each method (10min in Table 1).
>
> Table 7 in the appendix provides the number of generated plans for Best-of-N. We reproduce it below (mean ± std. dev. over 1k test problems, 10min budget):
>
> || BW | Log | Lab | Sok |
> |-|-|-|-|-|
> | Num. Plans | 24768.9 ± 11816.0 | 7243.4 ± 14927.1 | 10953.1 ± 16137.1 | 2186.5 ± 4266.4 |
>
> However, note that the number of model queries is not a good metric of method quality since
>
> a) our main objective is to reduce plan length given a fixed compute budget, and
>
> b) some methods (like ours) use truncated rollouts (/partial plan), which results in significantly lower per-query cost compared to generating full plans.
>
> Here we also present the number of partial plans/rollouts for OCLGen (uniform) (for experiments in Table 1 within 10min time limit):
>
> || BW | Log | Lab | Sok |
> |-|-|-|-|-|
> | Num. Plans | 30167.02 ± 14052.26 | 29939.04 ± 40159.94 | 20047.53 ± 5439.91 | 3520.13 ± 1470.60 |
>
> Please note that these numbers should not be directly compared to Table 7 (Best-of-N) since ours uses partial instead of full rollouts. Moreover, OCLGen (and other search methods) spend additional compute on expanding nodes, computing heuristics, etc. Our results however confirm that these additional compute steps, compared to merely sampling N times from the model and taking the best, are worth the effort and lead to improved solutions in less time.

---

> > ### Author Rebuttal · Reviewer_7Gbc · 2026-04-01
> >
> > The answers to my two questions are clear and detailed enough. The greedy baselines are pretty good in terms of coverage, but not as good as the proposed search method in terms of plan quality. This makes me wonder: what's the relation between the proposed search algorithm using these "rollouts" and MCTS, provided that the rollouts get to the "goal" in general? I don't expect a crisp answer to this question, and my (positive) recommendation won't depend in it, but there may be something interesting to be said about this relation.

---

> > > ### Author Response · Authors · 2026-04-04
> > >
> > > Thank you for your positive reassurance of our work. Please consider our following answer:
> > >
> > > > Relation between the proposed search algorithm using "rollouts" and MCTS, provided that the rollouts get to the "goal" in general?
> > >
> > > Thank you for this follow-up question. The base model generates rollouts (i.e., plans or plan segments) which are often valid but low-quality in terms of plan length. In both our method (OCLGen) and the MCTS baseline, rollouts allow for fast candidate solution generation to find better-quality, i.e., shorter, plans. One key difference, however, is how these rollouts are used in OCLGen and MCTS to expand the tree/graph. Moreover, due to the different node selection schemes in the algorithms, the distribution of starting nodes from which rollouts are called also differs. We elaborate on both aspects below.
> > >
> > > **Node expansion:** In one MCTS iteration, the tree is traversed down from the root node using a selection policy (e.g., PUCT). Once a leaf node is reached, its child nodes are expanded, i.e., successor states are generated and added to the tree. MCTS then runs a rollout from that selected leaf node, which can be seen as an attempt to complete a candidate or partial solution from that node. If the candidate solution is better than the current best, the best plan is updated accordingly. The outcome of the rollout (i.e., returns) is then backpropagated to update the Q-values of the edges traversed by the selection policy from the root. OCLGen similarly uses rollouts to generate candidate or partial plans from nodes picked from depth-specific open lists. Crucially, however, after a rollout is generated, OCLGen adds all nodes along it to the corresponding depth-specific open lists. Moreover, we assess model confidence at each step of the rollout and further branch to generate child nodes whenever confidence falls below a certain threshold. Adding rollout nodes to the search graph, together with adaptive expansion along rollouts, allows OCLGen to efficiently explore the search space while only branching where genuine potential exists. Note that it is not obvious how a similar mechanism could be incorporated into MCTS, due to concerns about the soundness of visitation statistics and general compatibility (see the discussion in the reply to reviewer SDEF). In short, unlike MCTS, OCLGen uses rollouts not merely for candidate solution generation and value estimation, but also to quickly and efficiently expand the search frontier while making branching adaptive with respect to model confidence.
> > >
> > > **Rollout start node distribution:** As noted above, MCTS selects nodes for expansion by traversing the entire tree from the root using a selection policy. Our method, by contrast, selects nodes by first choosing a depth level and then selecting a node within the depth-specific list, leveraging the ranking through the $f$-values. Rollouts, i.e., candidate plan completions, are then run from the selected nodes. The selection policies between our method and MCTS differ entirely, leading to two different distributions of nodes from which rollouts are initiated. In particular, our method distributes rollout start nodes more evenly across the relevant problem length due to depth-specific open list selection. In other words, our method uses this powerful tool of "fast candidate solution generation with generative models" more efficiently by spreading its use more evenly across depth. Our ablations support this, as removing depth-wise selection leads to the largest drop in solution quality (see Table 4).
> > >
> > > **In summary:** OCLGen and MCTS both use rollouts to find higher-quality candidate solutions. Crucially, however, the expansion and selection strategies in OCLGen allow for a more efficient and evenly distributed use of this generative model capability across the entire problem length, which our experiments confirm.

---

### Official Review · Reviewer_JorU · 2026-03-13

**Soundness:** 3
**Presentation:** 3
**Significance:** 3
**Originality:** 3
**Overall Recommendation:** 4
**Confidence:** 4

**Summary:**

OCLGEN is proposed here as an anytime OCL-style search process, executed at test time by the autoregressive planning models, that utilizes a range of techniques: open lists that are partitioned based upon depth; truncated policy rollouts; confidence-based adaptive branching; and a distributional heuristic considering a lower percentile of provided predicted cost to complete. OCLGEN's feasibility is evaluated utilizing domain specific GPT-2 policy models and BERT heuristic models within four domains: Blocksworld, Logistics, Labyrinth, and Sokoban; for each domain, OCLGEN was provided training data across 10^5 instances and provided separate held-out test data across 1,000 instances with a total inference budget of 10 minutes. Overall, OCLGEN produced plans with either 99.7% or 100% completion and produced shorter plans than MCTS and Best-of-N across each of the four domains; however, when examining the set of instances known to be optimal solutions, OCLGEN demonstrated that it solved significantly more optimal solutions than did MCTS in each of the four domains—e.g., for Blocksworld, OCLGEN solved 528 out of 630, whereas MCTS only solved 180 out of 630; for Labyrinth, OCLGEN solved 988 out of 1000, whereas MCTS only solved 605 out of 1000. Inside the final component of our analysis, we examined the components of OCLGEN and OCLGEN's self-improvement effect; we provided an example of OCLGEN solving for a total of 100% optimal solutions in Blocksworld and an example of OCLGEN solving for 94.7% optimal solutions after three rounds of development for the Sokoban domain.

**Compliance With Llm Reviewing Policy:**

Affirmed.

**Final Justification:**

Based on initial comments and rebuttal.

**Key Questions For Authors:**

- Report normalized inference costs (number of policy forward passes, heuristic forward passes, number of expanded states, average branching factors/rollout depth) across all methods. A yes answer further supports their hypothesis about efficiency.
- How sensitive are the results to tau_conf, rollout length, and heuristic percentile k, considering that tau_conf for Logistics is τConf = .2 while the other domains and τConf = .95? Robustness would improve my confidence in the method not being heavily tuned to any one domain.
- In the known-optimal subset, compare heuristic predictions to the true optimal cost-to-go instead of just suboptimal labels. This provides direct support to their bias-correction hypothesis for the percent threshold estimator.
- Conduct an OOD study using larger object counts/grid sizes or a generator/domain transfer experiment, even if you achieve small results, these will greatly increase the strength of their overall significance.

**Limitations:**

Somewhat. The authors do acknowledge future research opportunities (size OOD generalisation, transfer to new domains, self-improvement dynamics, pre-training where no initial solutions exist).

**Strengths And Weaknesses:**

Strengths:
- The paper offers a clear algorithm contribution because it recognizes an actual issue with root-centered Monte Carlo Tree Search (MCTS) in long-range planning, and it clearly provides a principled adaptation of the OCL search for generative planning.
- Evaluation of OCLGEN in distribution is very solid, with four domains being tested using 1000 test instances each; there are multiple comparisons to baselines, subsets of known optimal plans; there are also convergence plots; and multiple ablation studies have been conducted.
- The empirical results indicate a significant improvement consistently across domains based on plan length when MCTS and Best of N were close to 100% complete, and that MCTS has a substantially higher probability of finding an optimal solution compared to OCLGEN.
- The results of the self-improvement indicate that OCLGEN provides an improvement at the time of the decoding process, as well as providing an improvement to the planner as an operator when compared to MCTS.

Weaknesses.
- The technical justifications for their claims are mainly based on empirical evidence rather than theoretical explanations for expected behavior (e.g., depth partitioning, percentile heuristics) beyond the benchmarks provided.
- The paper does not normalize efficiency claims by compute, as they present wall-clock time instead of model-query counts, expanded states, or branching statistics, so it is difficult to determine how much of the performance gains result from each of these areas.
- Generalization of the work remains limited - data sets are domain-specific, training excludes instances that cannot be solved by FD-LAMA, and evaluation of results remains in the same generators and limited-size domains.
- The analysis of heuristics used by the authors is indirect because measurement of heuristic error is based on sub-optimal labels rather than true optimal cost-to-go, but the authors' primary claim is correcting the overestimation of the true optimal cost-to-go resulting from excessive reliance on suboptimal labelling.
- The cost of training is substantial and includes training separate domain-specific predictive/heuristic models on 4 independent compute instances, each with 8 A100 40GB GPUs.

---

> ### Author Rebuttal · Authors · 2026-03-30
>
> We thank the reviewer for their detailed comments. Below we address each concern, along with new experiments.
>
> # W1: Technical Justifications
>
> We agree that formal treatment strengthens the contribution and offer complementary perspectives:
>
> *Depth Partitioning.* If training plans are $\alpha>1$ times longer than optimal, a well-trained heuristic gives $h(n)\approx\alpha h^\*(n)$. For node $n_d$ at depth $d$ on an optimal path of length $L^\*$: $f(n_d)=d+\alpha(L^*-d)=\alpha L^\*-(\alpha-1)d$. Since $f$ decreases in $d$, search is biased toward deeper nodes and early mistakes are never corrected. Depth partitioning fixes this: within each level $d$ is fixed, so overestimation cancels in relative comparisons and $h$ remains a useful ranking signal.
>
> *Confidence-Based Expansion.* Exhaustive expansion explores $\sum_{i=1}^{d} b^i$ nodes to depth $d$. We follow the generative model when confident (branching factor 1), otherwise branch normally (factor $b$). If the model is confident at fraction $\gamma$ of decisions, the effective branching factor is $\tilde{b}=(1-\gamma)b+\gamma$, and explored nodes become $\sum_{i=1}^{d} \tilde{b}^i$. Even modest $\gamma$ yields large savings since the exponent amplifies reductions in $\tilde{b}$.
>
> *Percentile-Based Estimation.* A heuristic trained on suboptimal data shifts predictions above $h^\*(n)$. The true cost likely falls in the lower tail, motivating selection of a lower percentile. While not guaranteed for all distributions, we present example distributions in the appendix and compare against true optimal labels in Q3.
>
> # W2: Normalized Efficiency
>
> Note that our setting is anytime planning, which continuously improves plans until a budget is exhausted. Metrics like expanded nodes do not capture how effectively an algorithm *improves* solutions over time, hence our primary metric of solution quality vs. wall-clock time (Figure 3, Table 1).
>
> Moreover, model calls are not comparable units: the generative model produces sequences of different length; cost scales with sequence length; and KV caching makes autoregressive generation cheaper per token than independent passes. All learning-based methods share the same model, code framework, hardware, and parallelization, making the wall-clock comparison fair. We will add complementary statistics (iterations, selection depth) in the appendix; see also our related answer Q2 to reviewer 7Gbc.
>
> # W3: Generalization
>
> Sec. 5.2 shows that OCLGen generates valid plans where FD-LAMA failed, and outperforms MCTS. Generalization beyond trained object counts is limited by the generative model's token vocabulary which is a limitation shared by all baselines, not specific to our inference method.
>
> Yet, we ran an additional experiment: training on Blocksworld with 3–25 blocks while excluding 11,12,18,19, then testing on those (interpolate) and 26–40 (extrapolate). The model (Best-of-N) solved 100% of interpolation instances but 0% of extrapolation (unseen object names). The results on extrapolation are not surprising since the inputs contain unseen object names/tokens (e.g. block31). After shuffling object names during training, extrapolation rose to 74.0%. OCLGen further improved it to 79.87%. We are happy to include a corresponding section in the paper.
>
> # W4: Heuristic Analysis
>
> Please see Q3 below.
>
> # W5: Compute Cost
> The training setup is shared across all ML baselines and not specific to our method. We focus on the inference method. OCLGen can actually reduce cost: (1) it finds better solutions faster (Figure 3), and (2) it reduces required self-improvement cycles (Sec. 5.4).
>
> # Q1: See W2
>
> # Q2: Hyperparam. Sensitivity
>
> We evaluated $\tau_\text{conf} \in$ {0.2, 0.5, 0.95 } and $k \in $ {3, 10} on a validation set. Results on the full test sets:
>
> For $\tau_\text{conf}$:
> | | BW (0.2) | BW (0.5) | BW (0.95) | Log (0.2) | Log (0.5) | Log (0.95) |
> |-|---------|---------|----------|----------|----------|-----------|
> | Compl. | 100 | 100 | 100 | 100 | 100 | 100 |
> | Len. | 44.26±21.88 | 44.19±21.93 | 43.88±21.35 | 155.83±109.68 | 155.76±109.72 | 155.95±109.84 |
>
> For $k$:
> | | BW (3) | BW (10) | Log (3) | Log (10) |
> |-|---------|----------|----------|-----------|
> | Compl. | 100 | 100 | 100 | 100 |
> | Len. | 43.88±21.35 | 44.05±21.53 | 155.84±109.65 | 155.83±109.68 |
>
> Performance remains nearly identical across settings, consistently outperforming baselines, confirming robustness.
>
> # Q3: Heuristic vs. Optimal Cost
>
> We compared heuristic predictions to true optimal labels (MAE $\pm$ std. error):
>
> | | BW | BW (Iter 3) | Log | Lab | Sok | Sok (Iter 3) |
> |-|----|-----------|-----|-----|-----|------------|
> | Mode | 8.68±0.24 | 0.44±0.02 | 3.41±0.09 | 2.32±0.10 | 5.28±0.17 | 3.05±0.11 |
> | Perc  | 4.07±0.12 | 0.44±0.02 | 3.07±0.08 |  1.07±0.04 | 3.31±0.12 | 2.51±0.08 |
>
> The percentile estimator consistently moves predictions closer to optimal cost-to-go, directly supporting our hypothesis. Self-improvement (Iter 3) further reduce error.
>
> # Q4: See W3

---

> > ### Author Rebuttal · Reviewer_JorU · 2026-04-03
> >
> > Thank you for the rebuttal.

---

> > > ### Author Response · Authors · 2026-04-04
> > >
> > > We are grateful for your review of our work and for your constructive feedback. We hope our responses have resolved your concerns and will be reflected in your final assessment.

---

### Official Review · Reviewer_SDEF · 2026-03-15

**Soundness:** 2
**Presentation:** 4
**Significance:** 3
**Originality:** 3
**Overall Recommendation:** 3
**Confidence:** 4

**Summary:**

This manuscript proposes a generative AI planning framework named OCLGEN, which replaces Monte Carlo Tree Search (MCTS) with Open-Closed List (OCL) algorithm during sampling. The main motivation is that MCTS tends to explore more close to the root node, leading to “wide-but-shallow trees” unsuitable for planning problems. OCLGEN further introduces multiple modifications to the OCL, including truncated rollouts, depth selection, and adaptive expansion. Experiments over multiple planning domains show its advantage on task completion rate and plan lengths. Ablation study further confirms the efficacy of the proposed OCL modifications.

**Compliance With Llm Reviewing Policy:**

Affirmed.

**Key Questions For Authors:**

- What is the performance of OCLGen based on vanilla OCL procedure (as in Sec. 4.1) without “depth selection” or "adaptive expansion”  tricks to forcefully encourage expansion? How does it compare to MCTS (full)?
- What is the performance of MCTS (full) with also “depth selection” during the node selection stage, as well as “adaptive expansion” during rollouts? How does it compare to OCLGen?

**Limitations:**

Yes

**Strengths And Weaknesses:**

Strengths:
- Balancing exploration and exploitation during planning is an important problem to study.
- Analysis of the existing approaches and the motivation of most of the designs in the paper is clear.
- The paper is well-written.
- Experiments are thorough.

Weaknesses:
- While the motivation to improve exploration in the deeper layers is clear and reasonable, it is unclear why OCL is adopted over the MCTS. In the introduction section, the paper motivates the use of OCL as it “selects nodes globally”, which, based on my understanding, alleviates the bias of exploration over search depth in MCTS. However, the proposed OCLGen still relies on heuristic modifications such as “depth selection” and “adaptive expansion” to balance exploration over depth.
- Hence, it is unclear what is the benefit of using OCL, which is the foundation of the proposed method, especially since MCTS (full) is already performing at 100% completion rate in all testing domains in Table 1. While OCLGen achieves shorter plan lengths, it is unclear if it is contributed by the OCL or the modifications, which seem to be equally applicable in MCTS.

---

> ### Author Rebuttal · Authors · 2026-03-29
>
> Thank you for your thoughtful review. We address each concern below and provide additional experiments.
> # W1: Why OCL over MCTS?
> OCL and MCTS are based on fundamentally different principles. OCL selects nodes from a global open list. MCTS, by contrast, selects nodes locally by traversing the tree from the root via a selection policy (e.g. PUCT) until a leaf is reached, then backpropagates values to the root. This root-to-leaf traversal with backpropagation is the defining principle of MCTS. We argue that our modifications integrate naturally into OCL but not into MCTS, as detailed below.
>
> **Depth Selection** Depth-based selection chooses nodes from the depth-specific open lists, which more naturally fits into the OCL ("Open-Closed List") framework. Integrating this into MCTS without effectively switching to OCL raises fundamental questions: (1) Which scoring function selects within a depth bucket (value model? backpropagated Q-values? PUCT?)? (2) Do we expand the chosen node directly or start tree traversal from it? (3) Do we backpropagate to the root or only to the selected depth? Backpropagating beyond the selected depth breaks visitation statistics and undermines PUCT's foundations; backpropagating only to the selected depth creates independent sub-trees. Any sound resolution likely removes defining aspects of MCTS. E.g., removing tree traversal entirely and directly selecting expansion nodes from a set of open lists essentially leads to a method more similar to OCL.
>
> Despite the above concerns, we implemented a "depth-selection" MCTS where the selection phase (1) samples a depth level uniformly, (2) selects a node within that bucket via Q-values (initialized by the value model), (3) runs tree traversal from that node, and (4) backpropagates to the root. This variant overall performs worse  (except for Labyrinth) than baseline MCTS as shown in the table below (± std. dev.).
>
> ||BW|Log|Lab|Sok|
> |-|-|-|-|-|
> | Compl. | 100 | 100 | 100 | 99.6 |
> | Plan Len. | 56.78 ± 30.39 | 159.26 ± 111.54 | 15.17 ± 6.4 | 131.53 ± 101.78 |
>
> This might be due to this version breaking node visitation and Q-value statistics essential for the selection and backpropagation mechanisms in MCTS. Compared to OCLGen (Table 1 in paper), ours achieves substantially shorter plans across all domains.
>
> **Adaptive Expansion** In MCTS, new nodes are added during leaf node expansion, not rollouts. Rollouts produce Monte Carlo return estimates (and candidate plan solutions) but do not directly add new nodes to the tree. Adding intermediate rollout states to the tree raises unresolved questions: How are these nodes connected to the tree so the selection policy can reach them? How are visitation/Q-value statistics updated? In OCL, this is trivial: new nodes are simply appended to the global open list.
>
> In summary, both modifications are easily implemented using open list node access/selection, which is native to OCL but foreign to MCTS. Forcing them into MCTS either breaks soundness or transforms it into an OCL algorithm.
>
> # W2: Benefit of OCL given MCTS achieves 100% completion
> As stated in our introduction, the focus of this work is solution quality (plan length/optimality), not merely finding any solution. This is a substantially harder problem—e.g., optimal Blocksworld planning is NP-hard, while finding any solution admits polynomial-time algorithms [1]. Since training data is often suboptimal (collecting large-scale optimal plans is computationally infeasible), test-time search methods that improve plan quality are critical both for inference and model self-improvement. In our paper, we report completion rates to verify our method maintains solvability, but **the key metrics are plan length and optimality rate**, where OCLGen significantly outperforms all baselines. Please also see our answer to reviewer 4pj7, which demonstrates the statistical significance of our results.
>
> # Q1: Performance without depth select. and adaptive exp.?
> Please see Table 4 + Sec. 5.3 in the paper which ablates each modification individually. Here, we additionally ablate both together:
>
> ||BW|Log|Lab|Sok|
> |-|-|-|-|-|
> | Compl. | 100 | 100 | 100 | 99.6 |
> | Plan Len. | 49.9 ± 28.82 | 159.28 ± 111.4 | 13.06 ± 4.24 | 129.12 ± 99.42 |
>
> Removing both modifications degrades plan length as expected (+worse completion in Sok), confirming their importance.
>
> # Q2: MCTS with depth select. / adaptive exp. ?
> As argued above, cleanly integrating these modifications into MCTS is non-trivial and raises soundness concerns. Our best-effort implementation (see W1) performs worse than standard MCTS, further supporting that OCL is the more natural framework for these enhancements.
>
> We will gladly integrate the above discussion into the final version. Moreover, we are happy to address further questions and kindly ask the reviewer to reconsider their score in light of our responses.
>
> [1] Helmert, M. 2003. Complexity results for standard benchmark domains in planning. Artificial Intelligence

---

> > ### Author Rebuttal · Reviewer_SDEF · 2026-04-04
> >
> > Thanks for the rebuttal. Now my questions are addressed. I will increase the rating

---

> > > ### Author Response · Authors · 2026-04-04
> > >
> > > Thank you for reviewing our work and for the feedback. We are pleased our responses addressed your concerns and appreciate you updating your rating.

---

### Decision · Program_Chairs · 2026-04-30

**Decision:**

Accept (regular)

**Comment:**

Overall, the reviewers found that the paper makes a useful contribution by exploring the integration of an alternative test-time search approach with generative modeling. Although the empirical evaluation is limited to a moderate number of domains, the reported results show convincing improvements. More broadly, the paper helps open an interesting line of investigation on combining more advanced search methods with modern generative models, which is likely its clearest contribution.

The authors are encouraged to take the reviewers’ comments into account, especially in clarifying the distinction from prior search techniques used in similar settings.